# Transcription-dependent cohesin repositioning rewires chromatin loops in cellular senescence

Ioana Olan [1], Aled J. Parry [1,2,9], Stefan Schoenfelder [2,3,9], Masako Narita [1], Yoko Ito [1], Adelyne S. L. Chan[1], Guy St.C. Slater[1], Dóra Bihary [4], Masashige Bando[5], Katsuhiko Shirahige[5], Hiroshi Kimura [6], Shamith A. Samarajiwa [4], Peter Fraser [3,7✉] & Masashi Narita [1,8✉]

Senescence is a state of stable proliferative arrest, generally accompanied by the senescence-associated secretory phenotype, which modulates tissue homeostasis. Enhancer-promoter interactions, facilitated by chromatin loops, play a key role in gene regulation but their relevance in senescence remains elusive. Here, we use Hi-C to show that oncogenic RAS-induced senescence in human diploid fibroblasts is accompanied by extensive enhancer-promoter rewiring, which is closely connected with dynamic cohesin binding to the genome. We find de novo cohesin peaks often at the 3′ end of a subset of active genes. RAS-induced de novo cohesin peaks are transcription-dependent and enriched for senescence-associated genes, exemplified by *IL1B*, where de novo cohesin binding is involved in new loop formation. Similar *IL1B* induction with de novo cohesin appearance and new loop formation are observed in terminally differentiated macrophages, but not TNFα-treated cells. These results suggest that RAS-induced senescence represents a cell fate determination-like process characterised by a unique gene expression profile and 3D genome folding signature, mediated in part through cohesin redistribution on chromatin.

[1] Cancer Research UK Cambridge Institute, University of Cambridge, Robinson Way, Cambridge, UK. [2] Epigenetics Programme, The Babraham Institute, Babraham Research Campus, Cambridge, UK. [3] Nuclear Dynamics Programme, The Babraham Institute, Babraham Research Campus, Cambridge, UK. [4] MRC Cancer Unit, Hutchison/MRC Research Centre, University of Cambridge, Cambridge Biomedical Campus, Cambridge, UK. [5] Laboratory of Genome Structure and Function, Institute of Molecular and Cellular Biosciences, The University of Tokyo, Tokyo, Japan. [6] Cell Biology Centre, Institute of Innovative Research, Tokyo Institute of Technology, Yokohama, Japan. [7] Department of Biological Science, Florida State University, Tallahassee, FL, USA. [8] Tokyo Tech World Research Hub Initiative (WRHI), Institute of Innovative Research, Tokyo Institute of Technology, Yokohama, Japan. [9] These authors contributed equally: Aled J. Parry, Stefan Schoenfelder. ✉email: pfraser@bio.fsu.edu; masashi.narita@cruk.cam.ac.uk

Cellular senescence is a program of persistent cell-cycle exit, triggered by diverse stimuli, including genotoxic stress and excessive oncogenic signalling[1]. While senescence plays a role in limiting the propagation of damaged cells, persistence of senescent cells in vivo can have deleterious effects on tissue integrity[2,3]. Ample evidence suggests that this is primarily mediated through the senescence-associated secretory phenotype (SASP). While the composition of SASP can be different depending on cell types and senescence inducers, the major SASP factors, mostly identified in human diploid fibroblast models, include inflammatory cytokines and chemokines and extracellular matrix remodelling factors[3,4].

Senescence has been linked to dynamic alterations of the chromatin state, through the formation of senescence-associated heterochromatic foci (SAHFs), altered distributions of histone modifications and chromatin accessibility[2,5] or the appearance of new 'super-enhancers'[6]. In addition, the three-dimensional chromatin structure of senescent cells has been characterised using Hi-C technologies, mostly in terms of macro-domain structures, such as megabase-sized topologically associating domains (TADs) and larger-scale chromatin organization, A/B compartments, which represent 'open' (A) and 'closed' (B) chromatin states[7–11]. For example, the first study showed an increase in long-range interaction with a limited alteration in TAD borders during oncogene-induced senescence[7]. A/B compartments also appear to be stable[8], although some changes between the two compartments were also reported[9,10]. These studies have provided insights into how large-scale 3D chromatin landscapes can be altered during senescence. However, the relevance of the dynamic chromatin interaction in the gene regulation remains elusive.

The recent advance in Hi-C technologies and high-resolution Hi-C data have revealed additional structural units[12], such as chromatin loops[13,14] and enhancer-promoter (EP) contacts[15,16]. Vast majority of loop anchors (loop ends) are bound by CCCTC binding factor (CTCF) and the cohesin complex, which consists of three core subunits (SMC1, SMC3 and RAD21)[13,14]. It is thought that loops are a structural unit of gene regulation and enhancers and promoters within same loop domains are more likely to bind[12,17]. EP interactions can be dynamically altered during cell differentiation[16,18], playing a key role in cell type-specific gene expression. It has been shown that regulation of acute stress-responsive genes can be achieved through transcription factor (TF) recruitment to largely pre-existing EP contacts, suggesting that perturbation-responsive gene expression can occur under conserved EP networks in given cell types[15,19,20]. However, whether or not a similar mechanism is employed during senescence is unknown.

Here we characterize high-order chromatin structure alteration during oncogenic RAS-induced senescence (RIS), integrating Hi-C and high-resolution capture Hi-C (cHi-C) for selected regions, including senescence-relevant genes. We particularly focus on the dynamic nature of chromatin loops and EP contacts during senescence and highlight the extensive alteration of loop structures, which correlates with differential binding of cohesin, but not CTCF. Decreased cohesin binding is enriched around loop anchors. In contrast, de novo cohesin peaks in RIS cells are mostly associated with highly active genes within loop domains in a transcription-dependent and CTCF-independent manner. These de novo cohesin peaks tend to interact with neighbouring cohesin peaks, suggesting they might contribute to the new loop formation, whereby affecting local EP contact, in RIS cells. *IL1B* can be induced in both TNFα-treated and RIS human fibroblasts, and yet cohesin accumulation on the *IL1B* gene accompanied by new loop formation and EP alterations at the *IL1* locus are only observed in the RIS condition. Our data suggest that senescence can activate fundamentally different gene regulatory machinery from non-senescent stress responses.

## Results

### RAS-induced senescence exhibits significant interaction changes within and between TADs.

To study gene regulatory mechanisms in the 3D chromatin context at high resolution, we performed in situ Hi-C experiments as well as capture Hi-C (cHi-C) in normal growing and oncogenic *HRAS-G12V*-induced senescent (RIS) IMR90 human diploid fibroblasts (HDFs), using the 4-hydroxytamoxifen (4OHT)-inducible oestrogen receptor (*ER*) *HRAS* fusion system (*ER:HRAS^G12V*)[21]. Matching cHi-C libraries were generated for 62 selected genomic regions of interest (Supplementary Data 1). Growing and RIS Hi-C libraries yielded a total of 523 and 286 million valid reads, respectively, after removal of artefacts and duplicates (Supplementary Data 2). There was good agreement between biological replicates, as determined with HiC-Spector[22] as well as with HiCRep[23] (Supplementary Fig. 1a, b).

Using the Hi-C data, we identified 3488 and 3535 TADs in growing and RIS conditions, respectively, at 40 kb resolution. In agreement with a previous study[7], TAD borders were similar between conditions (estimated Normalized Mutual Information 0.98 between the sets of exact growing and RIS borders). We also found virtually no differences in the distribution of A/B compartments (Supplementary Fig. 1c), which were determined at 100 kb resolution[24]. A/B compartment score positively correlated with H3K27ac and H3K4me1 and negatively correlated with H3K9me3 ChIP-seq signal (Supplementary Fig. 1d). Gene expression and epigenetic information used to complement our Hi-C data were obtained from our previous studies using the same RIS IMR90 cell model[25–27]. We next estimated significant differential interactions between conditions with diffHic[28] using all the available replicates and found extensive alterations in chromatin contacts during RIS within TADs and between distal TADs (Fig. 1a, b and Supplementary Fig. 1e), similar to the previous oncogene-induced senescence study[7]. 2645 of the 3488 TADs (defined in growing condition) exhibited significant interaction changes, out of which 1621 were A compartment TADs (Supplementary Fig. 1f). We ranked the TADs by the number of significant changes occurring within them (Supplementary Fig. 1f, g and Supplementary Data 3). The most extensive interaction change occurred at the location of the *NRG1* gene (Fig. 1b and Supplementary Fig. 1g), which was reported as a senescence marker[29] and strongly upregulated during RIS (FDR 1.83e−315, log-fold change 4.56). The *NRG1* gene was largely (except for a few isoform-specific 5′ exons) encompassed in a H3K27me3-dense TAD in growing cells. However, the interactions within the gene body and with the nearby regions were almost entirely lost in RIS cells. As illustrated in our 3D TADbit[30] modelling (see "Methods"), the data suggest that the *NRG1* gene body is released from the heterochromatic TAD (Fig. 1c). This was accompanied by a significant increase in chromatin accessibility across the gene, as determined by differential binding analysis of growing and RIS ATAC-seq (Fig. 1b, bottom). Similar behaviour was observed within the second most changing TAD, which encompasses the *HMGA2* gene, encoding a regulator of senescence-associated heterochromatic foci[31] (SAHFs) (Supplementary Fig. 1g). The reduced contacts between *HMGA2* and neighbouring H3K27me3 (Supplementary Fig. 2a, b) were accompanied by upregulation of *HMGA2* during RIS in RNA-seq (FDR 2.275e−08, log-fold change 0.7) as well as qPCR analysis (Supplementary Fig. 2a, c). Genome-wide, we identified 102 upregulated genes dissociating from H3K27me3 regions during RIS (Supplementary Data 4), by subsetting all the significant

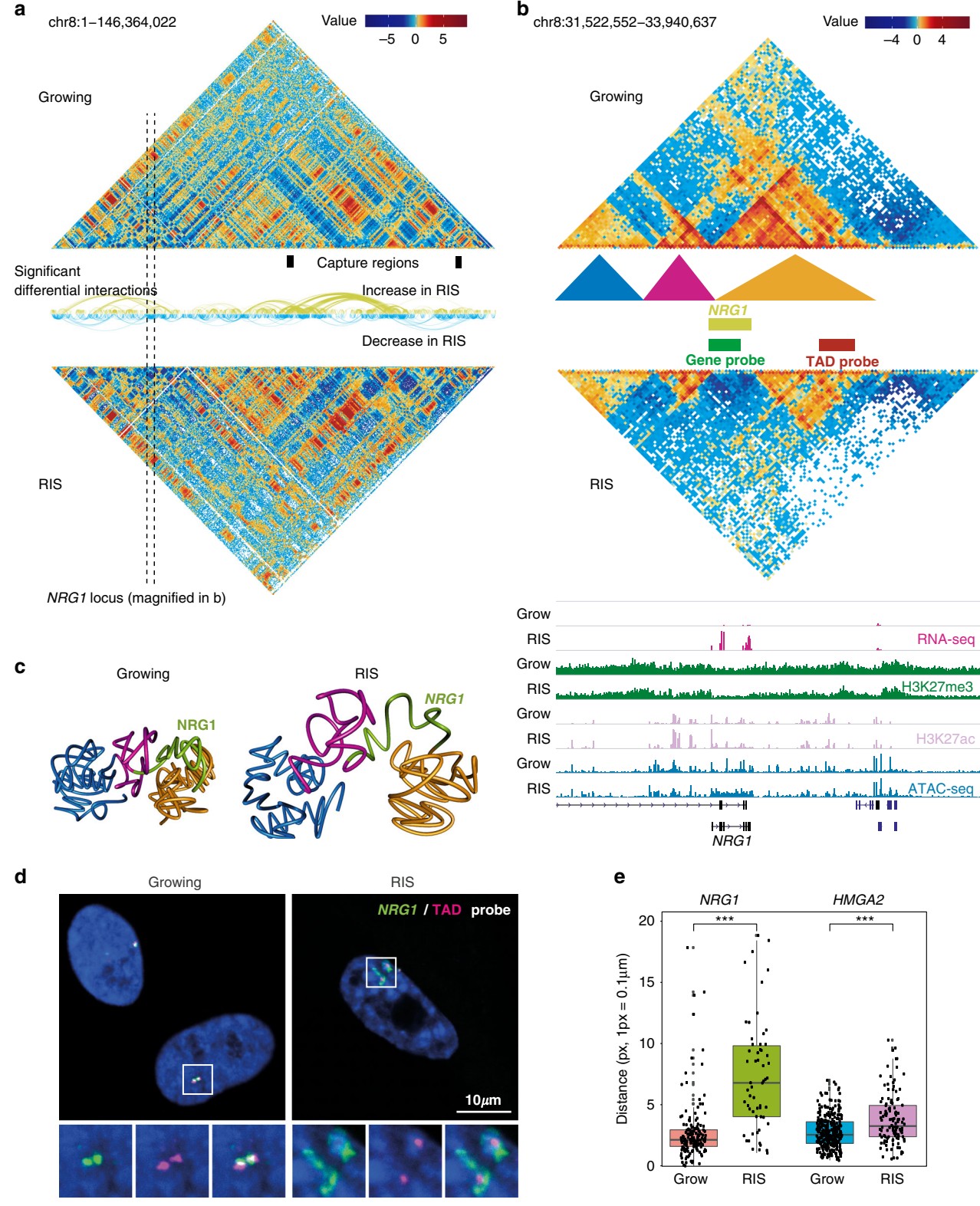

interaction changes for the ones occurring between genes and H3K27me3 regions. To validate the chromatin structural alteration of the two representative genes, *NRG1* and *HMGA2*, we performed DNA-FISH using BAC probes (probe size is ~155 kb) for gene bodies (gene probes) and the other end of the respective TADs (TAD probes) as depicted in Fig. 1b and Supplementary Fig. 2a. Consistent with our model, the average distance between the gene and TAD probes was very small, often

exhibiting nearly overlapping FISH signals in growing cells, but was significantly longer in RIS cells (Fig. 1d, e and Supplementary Fig. 2d). Note, in RIS cells, gene probes often showed a decondensed form, while TAD probes remained condensed (Fig. 1d and Supplementary Fig. 2d). These data suggest that H3K27me3 regions might contribute to long-range silencing of neighbouring genes through 3D positioning within TADs. This is consistent with a previous study, showing a similar distant gene

**Fig. 1 Changes in chromatin interactions during RIS. a** Hi-C matrices (300 kb resolution) of chromosome 8 in Growing and RIS cells (all available replicates were aggregated per condition as described in the Data visualization methods section); arcs represent significant interaction changes (100 kb resolution, determined genome-wide with only chromosome 8 shown here as an example); black boxes represent captured regions. **b** Hi-C interaction matrices at the *NRG1* locus, as marked by the dotted lines in (**a**), at 20 kb resolution, with TADs represented by coloured triangles (called at 40 kb resolution), as well as matching tracks for RNA-seq, ChIP-seq, and ATAC-seq (normalized and input-subtracted using THOR[62]). **c** Three-dimensional interaction modelling with TADbit at the *NRG1* locus in Growing and RIS, including *NRG1* (green) and surrounding TADs marked in (**b**). **d** Representative DNA-FISH images in growing and RIS cells, using the FISH probes corresponding to the *NRG1* locus (gene-probe, green) and nearby H3K27me3 region within the same TAD (TAD-probe, magenta), as marked in (**b**). Regions indicated by rectangles are magnified, showing two gene-TAD pairs in each condition. **e** Quantification of the average distance per cell between the gene-probe and the TAD-probe in growing and RIS cells, where each dot corresponds to one cell ($n = 159, 58, 321,$ and 116, respectively, from left to right, cells examined in two experiments consisting of growing and RIS conditions). Significance testing was performed using two-sided $t$-tests: ***$p \leq 0.001$ (left: $p = 3.721e{-}10$, right: $p = 9.195e{-}07$). Box plots correspond to the median, 25th to 75th percentiles, and the whiskers correspond to the 10th to 90th percentiles.

silencing mediated by H3K27me3 interactions in embryonic stem cells[32].

**Enhancer-promoter interactions are extensively altered during RAS-induced senescence.** We next focused on gene expression and its association with regulatory elements. We annotated the genome-wide interaction changes between promoters and active enhancers, focusing on the enhancer-promoter interactions of genes differentially expressed during RIS. Enhancers (Supplementary Fig. 3a) were defined as regions marked by H3K27ac, H3K4me1, and ATAC-seq peaks, which do not overlap gene promoters, similarly to Tasdemir et al.[6] (details in "Methods"). Promoters (Supplementary Fig. 3b) were defined as 5 kb regions around transcription start sites (TSS, gene definition from GENCODE19). We first used the high resolution ('*HindIII* resolution', median 4 kb) cHi-C data to identify any differential EP pairs. Within the captured regions (Supplementary Data 1), we identified 870 EP pairs that showed significantly altered interactions during RIS, involving 149 differentially expressed genes (Supplementary Fig. 4a).

To gain a genome-wide picture, we next analysed Hi-C data and identified 15,618 'EP interactions', which significantly changed at 100 kb resolution. However, these EP contacts are likely to contain many false positives due to the large bin sizes compared to average enhancer or promoter size. To increase the accuracy of this estimate, we developed a strategy to filter the Hi-C EP interactions by minimising the EP changes annotated in Hi-C and not in cHi-C over captured regions (likely to be false positives), while maximising the EP changes annotated both in Hi-C and cHi-C (Supplementary Fig. 4b): enhancers with sizes greater than 7.5 kb and bin sizes smaller than 30 kb fulfilled these conditions. Using these filters, we identified 719 EP changes genome-wide from Hi-C data, involving 553 differentially expressed genes (Supplementary Fig. 5a). Combining Hi-C and cHi-C analyses, we identified 1004 confident EP differential interactions in total (Supplementary Data 5).

The distances between interacting enhancers and promoters from both Hi-C and cHi-C were below 2 Mb, consistent with the previous studies[15]. The EP network determined using cHi-C showed structures with a wide range of complexity, likely due to the high-resolution interaction information, consisting of 79 components with up to 15 nodes (enhancers or promoters) (Supplementary Fig. 4a). The complex rewiring was exemplified by the *IL1* and *MMP* loci, which include major SASP genes (Fig. 2a and Supplementary Fig. 4a). Although the Hi-C EP network, consisting of 479 components, was more disconnected and mostly represented a single EP interaction, the largest component consisted of 13 enhancers differentially interacting with the *INHBA* gene promoter (Supplementary Fig. 5a, chromosome 7). Of note, *INHBA* encodes a SASP factor which has been previously linked to super-enhancer activation in RIS

cells[6]. In terms of directionality, differences in H3K27ac binding over the enhancer in an EP interaction pair were positively correlated with the log-fold change of the interaction (0.24, $p$-value 9.353e−15) as well as with the gene expression log-fold change (0.14, 6.383e−6).

Gene set enrichment analysis (Supplementary Fig. 5b) using genes involved in differential EP interactions, which constitute 15% of all significantly differentially expressed genes in RIS, showed that transcriptionally upregulated genes (in RIS) were significantly enriched for 'inflammatory' terms, whereas the downregulated genes were enriched for 'cell cycle' terms. While further experimental validation is required, our data suggest that the two senescence hallmarks, the SASP and proliferative arrest, might be controlled through the rewiring of the EP network.

**New loop formation is observed at the *IL1* locus in RAS-induced senescence, but not with TNFα treatment.** The '*IL1* cluster', which was captured in our cHi-C libraries, encompasses the *IL1* ancestral family[33] (including *IL1A*, *IL1B*) and several other genes (such as *CKAP2L*) on chr2q13. Both *IL1A* and *IL1B* encode key proximal SASP components, which are integral parts of SASP regulation[34,35]. The localization of *CKAP2L* (encoding a mitotic spindle protein) within the *IL1* cluster is highly conserved and the expression of *CKAP2L* is tightly controlled during the cell cycle[36]. Our cHi-C showed dynamic sharing of enhancers between *IL1A, IL1B*, and *CKAP2L* during RIS. The differential interaction matrix of cHi-C at the *IL1* locus showed new loop formation, compared with loops that previously defined by high-resolution Hi-C in IMR90 cells[13], segregating *IL1A* and *CKAP2L* from *IL1B* and therefore increasing the specificity of their enhancer-associations. Consistently, *IL1A* and *IL1B* began to interact more frequently with enhancers located within their respective new loops in RIS (Fig. 2e). Moreover, *CKAP2L*, which was downregulated during RIS, interacted less frequently with the same downstream enhancers that *IL1B* began to contact more frequently (Fig. 2e). The data indicate that increased new loop formation and segregation of EP interactions occur at this locus, suggesting new loop formation around the *IL1B* gene.

This finding is in marked contrast to the *IL1* induction in a TNFα acute inflammatory scenario, in which gene regulation can be achieved without any detectable alteration to the EP landscape. Using high-resolution (5–10 kb) Hi-C maps, Jin et al.[15] have shown that a transient TNFα treatment of IMR90 cells leads to upregulation of *IL1A* and *IL1B* with increased binding of NF-κB (a major inflammatory TF) to active enhancers of its targets. In addition, *IL1A* and *CKAP2L* were shown to be induced simultaneously via shared enhancer binding. The authors concluded that gene expression alterations mostly occur via TF binding to 'pre-existing' EP complexes, at least upon TNFα treatment[15]. We re-analysed the Hi-C data from this study using the analysis pipelines described in this study and, like Jin et al.,

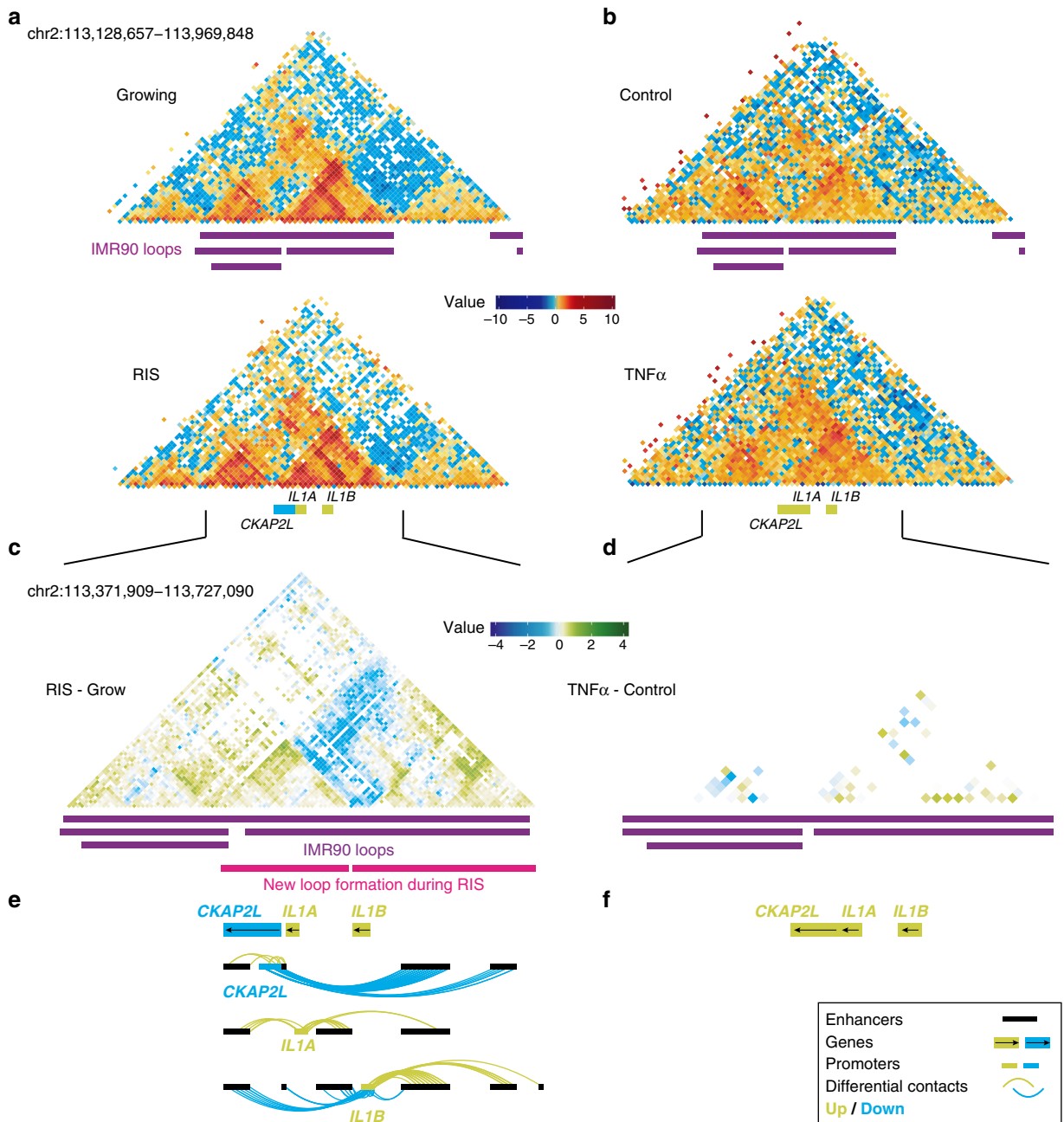

**Fig. 2 Reorganization of the local chromatin neighbourhood at the _IL1_ locus. a**, **b** Hi-C matrices (10 kb resolution) of the _IL1_ locus corresponding to growing and RIS IMR90 cells (**a**) and control and TNFα-treated IMR90 cells from Jin et al.[15] (**b**). **c** Differential capture Hi-C matrix at the _IL1_ locus (log-fold change of RIS/growing interactions estimated genome-wide using all replicates) at _HindIII_ resolution, with annotated growing IMR90 loops (from Rao et al.[13]) and inferred new loop formation in RIS cells. **d** Differential Hi-C matrix at the _IL1_ locus (log-fold change TNFα) at 10 kb resolution. **e**, **f** Significant differential enhancer-promoter contacts between promoters of differentially expressed genes at the _IL1_ locus and associated enhancers, aligned with (**c**) and (**d**), respectively. .

did not observe any significant changes upon TNFα treatment (Fig. 2b, d, f). This reveals a fundamentally distinct mechanism for the induction of inflammatory cytokines during senescence and acute inflammation. The anti-correlation between _IL1_ and _CKAP2L_ expression with significant EP interaction alterations was observed during RIS, but not with TNFα treatment, implying a senescence-specific decoupling mechanism within an otherwise co-regulated locus encoding key cytokines and cell cycle genes.

**Correlation between cohesin binding changes and loop reorganization during RAS-induced senescence.** To investigate

potential mechanisms underlying the observed EP changes during RIS, we generated ChIP-seq data for CTCF and cohesin (RAD21 and SMC3), chromatin structural proteins associated with chromatin loops[13,14], in both growing and RIS IMR90 cells. We found 44,764 and 53,563 CTCF peaks in growing and RIS cells, respectively. Comparative analysis identified 1774 CTCF peaks that were significantly altered during RIS. 96% of the CTCF changes were associated with increased binding in RIS (Fig. 3a). In contrast, RAD21 binding, represented by 26,374 and 24,355 peaks in growing and RIS, respectively, changed significantly at 4553 sites, of which 81% corresponded to decreased binding (Fig. 3a). Similar results were obtained for SMC3 ChIP-seq, which

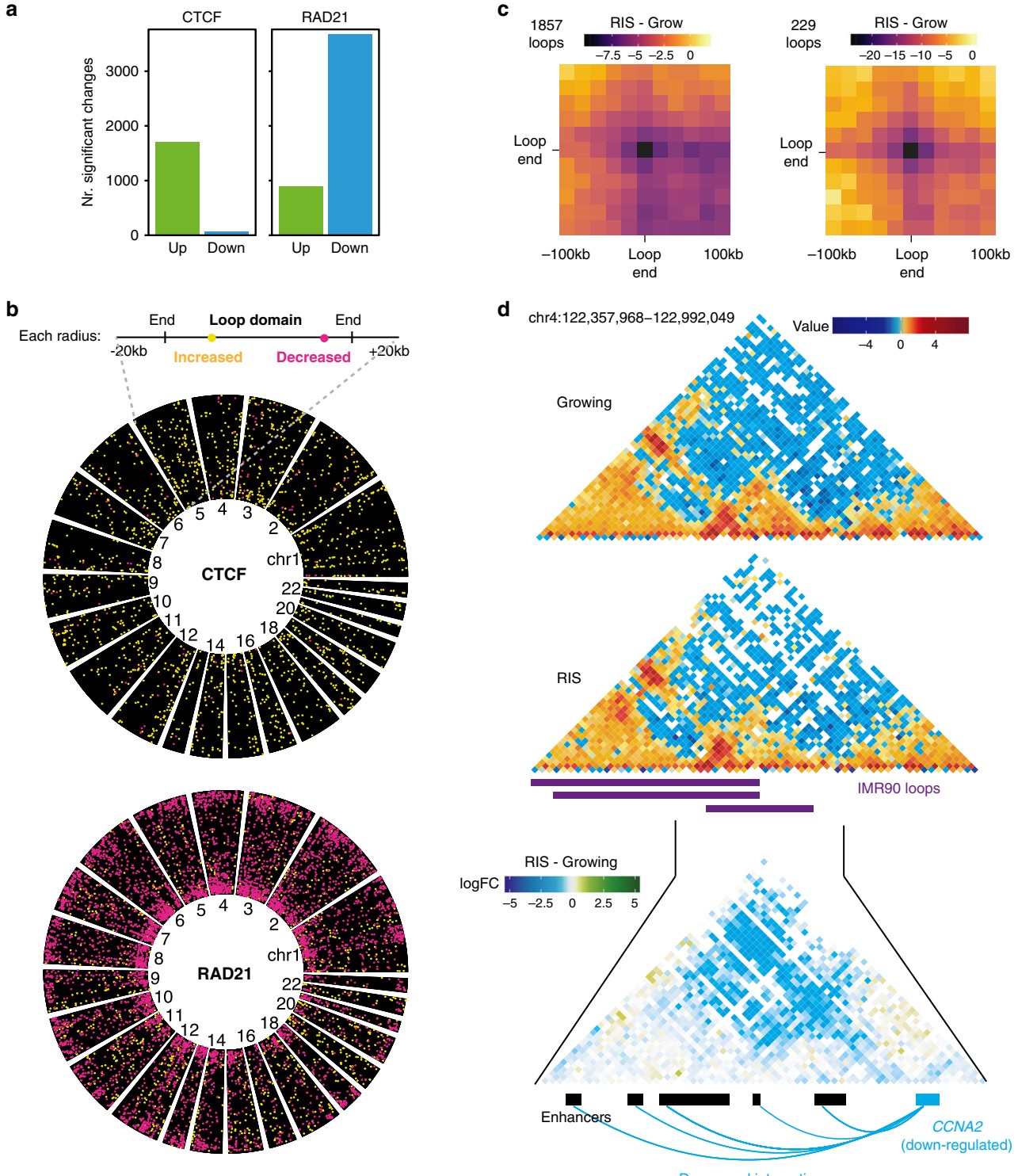

**Fig. 3 Correlation between cohesin redistribution and loop rewiring during RIS. a** Number of CTCF and cohesin ChIP-seq peaks with increased (green) and decreased (blue) binding in RIS relative to growing (FDR 0.05, differential binding analysis performed with THOR). **b** Position of CTCF and cohesin binding changes relative to the growing IMR90 loop spans (loops annotated in IMR90 cells by Rao et al.[13]); each loop is represented as a radial segment linking the two loop anchors with an extra 20 kb at each end. Each significant binding change is represented as a yellow (increased) or magenta (decreased) dot, for both CTCF and cohesin. **c** Differential aggregated Hi-C interactions neighbourhoods (20 kb resolution) of all IMR90 loops (1857 loops) that exhibited significantly decreased cohesin binding at one or both loop ends (left). Compare to the subset (229 loops) with significantly decreased Hi-C interactions during RIS (right). **d** Growing and RIS Hi-C matrices (10 kb resolution) centred on the IMR90 loop (from Rao et al.[13]) consisting of the *CCNA2* gene promoter and associated enhancers, as well as the cHi-C differential log-fold change matrix (5 kb resolution) of this loop. Significantly decreased interactions between the *CCNA2* gene promoter and associated enhancers are shown as blue arcs.

correlates well with RAD21 ChIP-seq signal (Supplementary Fig. 6a). Thus, although substantial numbers of peaks were gained in both CTCF and cohesin, a large fraction of cohesin binding was diminished.

Next, we investigated where the CTCF and cohesin binding changes occurred with regards to genomic features and loops (the latter previously defined in normal IMR90 cells by Rao et al.[13], Fig. 3b, see "Methods"). First, we checked whether the loops[13] were indeed enriched for CTCF and cohesin binding in our data. 90% of the loop anchors were bound by both cohesin and CTCF within 10 kb of the anchor. 3154 (out of 7647) loops showed changes in either CTCF or cohesin binding. Most of the loops correlated with cohesin changes exhibited cohesin losses at their ends (1857 loops), followed by loops with cohesin gains inside the loop (823 loops) (Supplementary Fig. 6b). The CTCF alterations (mostly increases as shown in Fig. 3a) occurred inside loops, rather than near loop ends (Fig. 3b). Such strong colocalization between cohesin loss and loop anchors suggests that extensive loop reorganization might occur during RIS, mostly through redistribution of cohesin rather than CTCF (Fig. 3b).

To visualize the relationship between cohesin reduction at loop ends and their physical contacts, we aggregated interaction neighbourhoods (at 20 kb resolution) centred on selected loop ends (Fig. 3c, Methods), a similar approach to the previously published method 'Aggregate Peak Analysis'[13]. This approach involved extracting all sub-matrices representing the interactions between the 200 kb regions centred on each loop end. This resulted in profiles of the interaction neighbourhoods of each loop, which were averaged in each of the two conditions, growing and RIS. We aggregated the interaction neighbourhoods of the loops with cohesin reduction at one or both ends (1857 loops) and found a trend of decreased interaction between loop ends in RIS compared to growing cells, suggesting a global correlation between the decreased cohesin binding at loop anchors and reduced loop formation (Fig. 3c, left). We next searched for the strongest loop reduction events by overlapping these 1857 loops with interactions which significantly changed during RIS based on our earlier genome-wide differential analysis at multiple resolutions, either from cHi-C or Hi-C, and identified 229 loops (Fig. 3c, right). In terms of enhancer-promoter interactions potentially affected by loop disruption, 430 differential EP contacts were nested within the 3154 loops described earlier (with cohesin and/or CTCF alterations either at the ends or inside the loops), involving 349 genes. An example of a decreased loop interaction affecting EP contacts was represented by the cell cycle regulator *CCNA2* (Fig. 3d).

**De novo cohesin binding at active genes including the *IL1* locus during RAS-induced senescence**. The vast majority of cohesin binding increases occurred de novo in RIS, compared to the decreased binding, which did not result in complete cohesin binding loss (Fig. 4a). The genes and enhancers studied in the *IL1* locus belong to the same loop identified in normal IMR90 cells (see Fig. 2c, e)[13]. This is consistent with enhancer sharing between these genes and their co-regulation in response to TNFα in these cells[15]. We found a de novo cohesin peak close to the 3′ end of *IL1B* in RIS cells, independent of CTCF binding (Fig. 4b). A similar cohesin peak was observed when RIS was induced via constitutive expression of HRAS^G12V without the ER-tag not only in IMR90, but also in WI38 HDFs (Fig. 4c). These data suggest that loop reorganization at the *IL1* locus might be associated with de novo cohesin binding (Fig. 2e). Importantly, we performed cohesin and CTCF ChIP-seq in IMR90 cells with or without TNFα treatment: increased cohesin occupancy or altered regulatory chromatin interactions at the 3′ end of *IL1B* were not

observed in response to TNFα treatment, where no new loops were detected, despite activation of NF-κB signalling and upregulation of *IL1A* and *IL1B* (Figs. 2f, 4d and Supplementary Fig. 7a, b). Additionally, we observed an increase in the contact intensity between the new cohesin peak and the anchors of the loop (Fig. 2c, e). Collectively, these data suggest that the de novo cohesin peak might contribute to the formation of new loops in the *IL1* locus and that within each loop domain, EP pairs might preferentially contact (Fig. 4e). Strikingly, the *MMP* locus, which contains other major SASP genes, was also characterized by the appearance of de novo cohesin at the 3′ end of *MMP1* (and, to a lesser extent, *MMP3*), as well as loop reorganization around the new cohesin peak (Supplementary Fig. 8a, b). We confirmed that the cohesin increases at this locus also occurred in RIS WI38 cells (Supplementary Fig. 8c).

The elongated shape of the cohesin peaks without CTCF binding at the 3′ end of *IL1B* and *MMP1* was reminiscent of recently reported[37] transcription-driven 'cohesin islands', which appear at the 3′ end of active convergent genes in double knockout (DKO) mouse embryonic fibroblasts (MEFs) of *Ctcf* and the cohesin release factor *Wings apart-like* (*Wapl*). The authors proposed that cohesin is loaded onto chromatin at the TSSs of a large number of active genes and is then relocated though transcription: if there is no CTCF in the way and no efficient cohesin release at the 3′ end of active genes, cohesin accumulates at the 3' end of these genes[37]. A similar pattern of cohesin binding has been reported in wild-type yeast, which lacks a CTCF equivalent[38–40]. Thus, we hypothesized that genes highly active in RIS somehow allow for the accumulation of cohesin at their 3' ends in a transcription-dependent manner, potentially promoting loop reorganization. To test this, we compared transcript abundance and cohesin binding at the 3' end of genes. Both convergent (genes on opposite strands that terminate in the same place) and isolated (no overlap with other genes) genes in RIS IMR90 cells exhibited cohesin island-like binding and cohesin binding positively correlated with gene expression (Fig. 4f and Supplementary Fig. 9a, b). Consistent with the lack of cohesin islands detected in wild-type MEFs[37], very few cohesin islands were detected in normal growing IMR90 cells. To confirm that cohesin islands were associated with the cellular condition, rather than a specific subset of active genes, we examined genes highly transcribed in both RIS and growing, but at higher levels in growing cells, for cohesin islands. Despite the reduced expression levels, cohesin islands were much more pronounced in the RIS condition (Supplementary Fig. 9c).

**'Cohesin island' formation during RAS-induced senescence is transcription-dependent**. To further investigate the transcriptional dependence of cohesin islands observed in RIS cells, we performed ChIP-seq experiments in RIS cells with or without 5,6-dichloro-1β-D-ribofuranosylbenzimidazole (DRB) treatment, a transcription elongation inhibitor. The de novo cohesin peaks at the *IL1B*, as well as *MMP1* sites, but not the sharp cohesin peaks colocalized with CTCF, disappeared completely (Fig. 5a and Supplementary Fig. 8d). To gain a global view, we defined genome-wide RIS-associated cohesin islands by performing differential binding analysis between the cohesin ChIP-seq libraries from RIS cells with and without DRB treatment. We found 574 wide cohesin peaks (between 2 and 20 kb wide), which were lost with DRB treatment (Fig. 5b). 531 islands were associated with 614 genes, not only near the gene ends, but also on gene bodies and promoters (Supplementary Fig. 9d). Note, we found a small number of cohesin islands (27 peaks) at enhancers with no obvious overlap with specific gene loci (Supplementary Fig. 9d). Whether or not these peaks are associated with specific

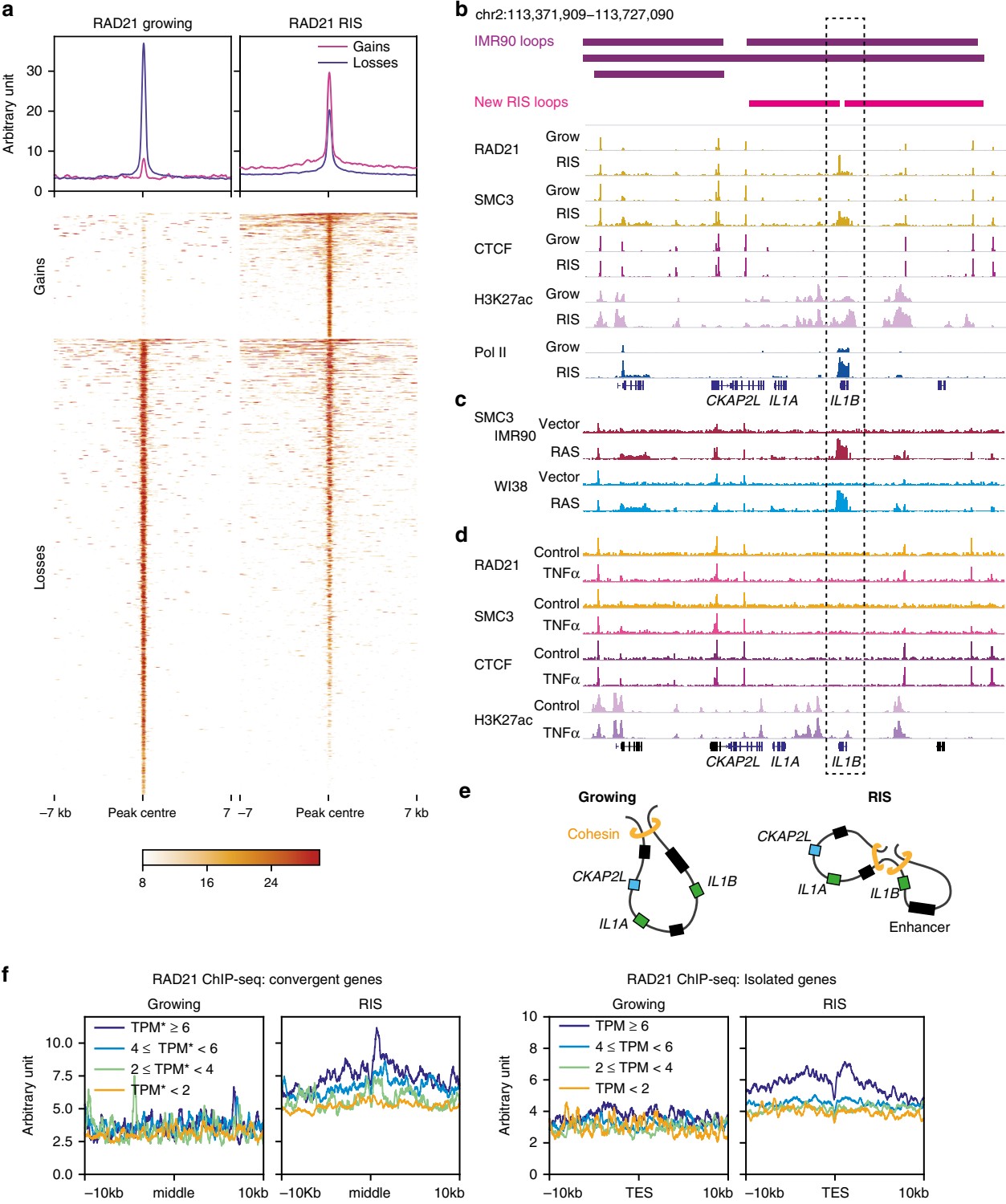

**Fig. 4 De novo cohesin binding is associated with high gene expression during RIS. a** Profiles and heatmaps of THOR-normalized differential RAD21 ChIP-seq signal in growing and RIS IMR90 cells. Differential signals ('Gains' or 'Losses' during RIS) were determined with THOR[62] at FDR 0.05. **b** Correlation between the inferred position of the new loop formation during RIS and indicated THOR-normalized ChIP-seq at the *IL1* locus. **c** THOR-normalized SMC3 ChIP-seq signal at the *IL1* locus in RIS IMR90 and WI38 cells, induced by constitutive expression of *HRAS^G12V*. Grow, matched controls, which expressed control vector. **d** THOR-normalized ChIP-seq signal of cohesin (RAD21 and SMC3), CTCF and H3K27ac at the *IL1* locus in TNFα-treated and matched control IMR90 cells. **e** Simplified model of new sub-loop formation within the loop encompassing the *IL1A*, *IL1B* and *CKAP2L* promoters, separating *IL1B* from *IL1A* and *CKAP2L*, along with their specific enhancers. **f** Cohesin (THOR-normalized RAD21 ChIP-seq signal) distribution at the 3' end of genes in RIS cells, grouped by log-transcripts-per-million (TPM) expression at convergent genes (overlapping extended 3' ends) and isolated genes (no overlap with other genes). middle, middle points between the converging 3' ends; TES, transcriptional end site. In the case of convergent genes, both genes in the pair were in the same expression category.

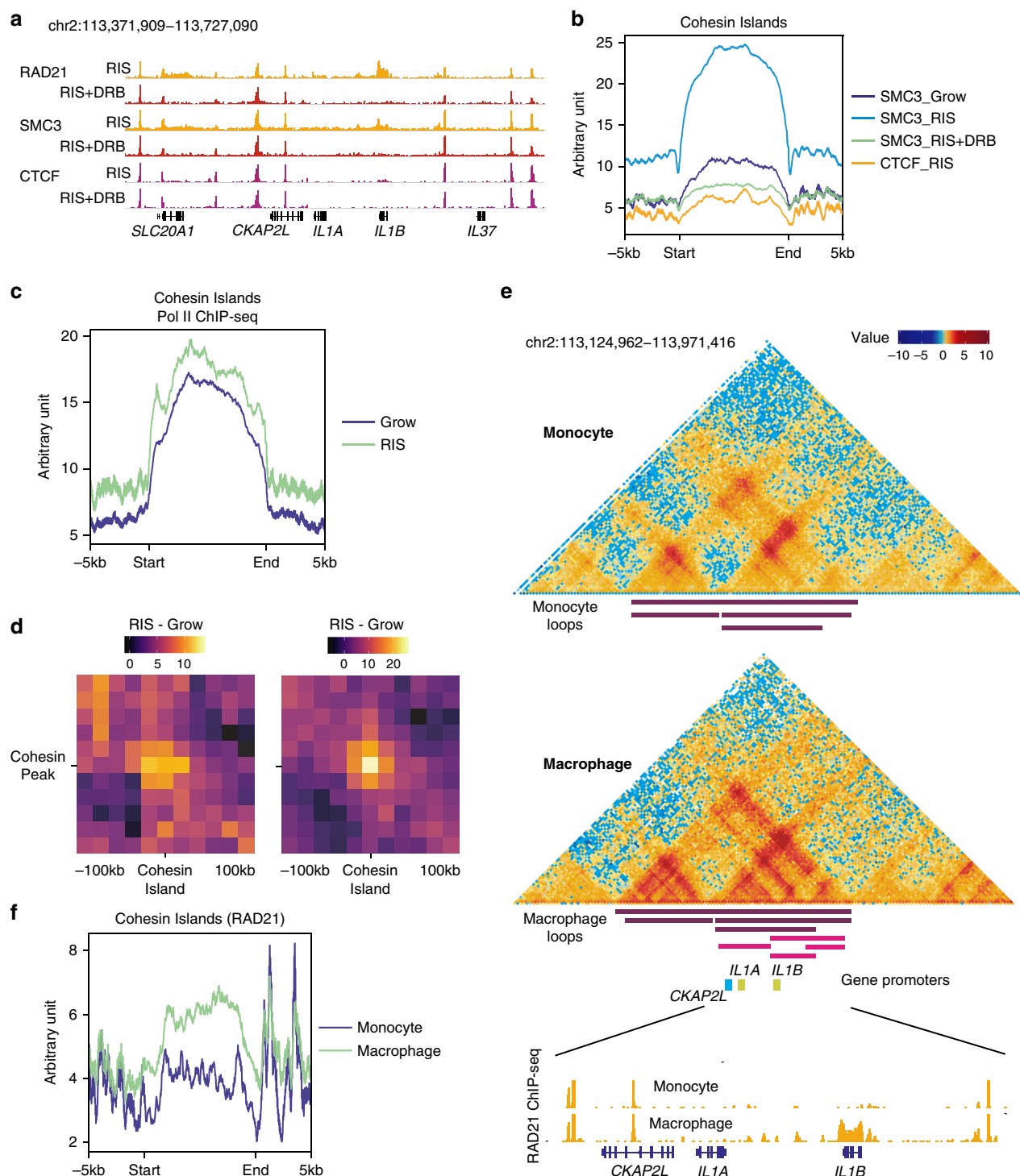

**Fig. 5 Transcription-dependent cohesin island formation during RIS and macrophage differentiation. a** Representative genome browser images of indicated THOR-normalized ChIP-seq at the *IL1* locus with and without DRB treatment in RIS IMR90 cells. **b** Averaged SMC3 and CTCF ChIP-seq signal in indicated conditions in IMR90 cells over all scaled cohesin islands identified, flanked by extra 5 kb regions. **c** Pol II ChIP-seq profile in growing and RIS over all cohesin islands, as in (**b**). **d** Differential aggregated interaction neighbourhoods (at 20 kb resolution) between RIS and in growing. The left panel represents all interactions between cohesin islands and nearby cohesin peaks within 150 kb of each other. Compare to the right panel, which represents only significantly increasing Hi-C interactions during RIS between each cohesin island and nearby cohesin peaks (within 250 kb either side). **e** Cohesin islands at *IL1B* during macrophage terminal differentiation and loop representations determined by Phanstiel et al.[44]. Reanalysis of Hi-C matrices (5 kb resolution) of THP-1 monocytes and PMA-induced macrophages from Phanstiel et al.[44] as well as RAD21 ChIP-seq from Heinz et al.[45] in the same cell context. **f** RAD21 ChIP-seq signals in monocyte and macrophage[45] over 65 cohesin islands shared by PMA-induced THP-1 macrophages and RIS IMR90 cells.

transcriptional activity is under investigation. The vast majority of those genes were highly expressed (89%) and in general, genes with cohesin islands had higher expression than genes without (Supplementary Fig. 10a). Moreover, similar to the previous study[37], 374 of those genes formed pairs of convergent genes. In addition to *IL1B* and *MMP1* (discussed above), the genes with cohesin islands were enriched in pathways that have been implicated in senescence, such as *Wnt*, Autophagy and NF-κB signalling[2,41,42] (Supplementary Fig. 10b), and included key SASP factors.

Consistent with the original cohesin islands defined in *Ctcf-Wapl* DKO MEFs[37] the lack of CTCF binding of cohesin islands in RIS cells appeared to a general trend (Fig. 5b). Constitutive cohesin peaks which overlap CTCF peaks were not affected by the DRB treatment (Supplementary Fig. 10c). Moreover, the same cohesin islands defined in RIS IMR90 cells also occurred in RIS WI38 cells, but not in TNFα-treated IMR90 cells (Supplementary Fig. 10d, e). Consistent with their association with active transcripts, the epigenetic profile of cohesin islands showed a global trend of increased chromatin accessibility (ATAC-seq) and histone modifications associated with active chromatin such as H3K27ac ChIP-seq (Supplementary Fig. 10f). In addition, we conducted RNA polymerase II (Pol II) ChIP-seq in growing and RIS IMR90 cells and found that cohesin islands, but not the CTCF-positive constitutive cohesin peaks, were enriched for Pol II binding (Figs. 4b, 5c and Supplementary Fig. 8b), further supporting the transcriptional dependency of RIS-associated cohesin islands.

**Cohesin islands show increased interactions with nearby cohesin peaks.** Next, we assessed whether RIS-associated cohesin islands modulate local chromatin structure, as observed at the *IL1* and *MMP* loci. Interactions between all (574) cohesin islands and surrounding cohesin peaks (4497 interactions) were increased in RIS in the Aggregate Peak Analysis (Fig. 5d, left). 185 cohesin islands exhibited significantly increased binding to local cohesin peaks (474 interactions, Fig. 5d, right). While experimental validation is necessary, these data suggest that cohesin islands likely contribute to changes in chromatin architecture during RIS, via de novo loop formation.

**Similar cohesin island formation is detected at the IL1 locus during macrophage differentiation.** Finally, we asked whether changes similar to those observed at the *IL1* locus in RIS cells occur in any other context. In most other cell types with cohesin information from ENCODE[43], the *IL1* locus cohesin binding pattern was similar to 'normal' IMR90 cells, suggesting that the loop structure at this locus is mostly conserved. However, Phanstiel et al.[44] recently reported *IL1B* upregulation and concomitant loop formation at the locus during terminal differentiation of monocytes into macrophages, another fate-determination process of lineage-committed cells. They generated high-resolution Hi-C maps and RNA-seq datasets in the human monocytic leukaemia cell line THP-1 both before (monocytes) and after (macrophages) phorbol myristate acetate (PMA) treatment. We reanalysed these datasets and found that the new loop formation in the *IL1* locus in THP-1 macrophages was similar to RIS (Fig. 5e), concomitant with similar expression changes: upregulation of *IL1A* and *IL1B* and down-regulation of *CKAP2L*. Reanalysing RAD21 ChIP-seq data in the same THP-1 cell model from Heinz et al.[45] revealed a de novo cohesin peak around *IL1B* (Fig. 5e). Genome-wide, cohesin binding also correlated with transcription levels (Supplementary Fig. 10g). 65 genes exhibited cohesin islands in both RIS and THP-1 macrophages (Fig. 5f). Together, these data suggest that transcription-dependent cohesin accumulation also occurs during macrophage

terminal differentiation and particularly, the same cohesin-mediated loop alteration at the *IL1* locus (as in RIS) might facilitate transcription of genes in this locus.

## Discussion

We show that significantly altered EP contacts, associated with gene expression changes, occur during RIS. This is in stark contrast to proinflammatory gene expression programs in response to acute stress or signalling cues, which appear to be predominantly driven by TF recruitment and remodelling of epigenetic chromatin signatures, rather than by dynamic alteration of EP interactions. Our data indicate that EP contacts in HDFs exhibit plasticity, being susceptible to further modulation towards senescence. EP contacts in lineage-committed cells also exhibit plasticity towards terminal differentiation[20,44]. Mechanistically, our data suggest that this can be at least partly explained by the formation of transcription-dependent cohesin islands. We also observed the induction of cohesin islands during macrophage terminal differentiation (Fig. 5e, f), suggesting that their formation is not solely linked to senescence.

Despite the common features, there are some differences between cohesin islands between RIS and *Ctcf-Wapl* DKO MEFs. Generally, cohesin islands tended to be wider than the CTCF-associated structural cohesin peaks, yet narrower in RIS HDFs and macrophages (up to 20 kb) than those in *Ctcf-Wapl* DKO MEFs[37] (up to 70 kb). Localization of RIS-associated cohesin islands appeared less restricted to 3'-ends of genes and they were often observed on gene bodies. The reasons behind these apparent differences are not clear but one possibility is the difference between the two genetic backgrounds. Further studies are required to understand how cohesin islands are formed in the presence of CTCF and WAPL in RIS fibroblasts and THP-1 macrophages. Importantly, in addition to their susceptibility to the DRB treatment, the RIS cohesin islands exhibited a strong correlation with Pol II ChIP-seq profiles, supporting the same mechanism, the transcription-dependent cohesin relocation, as proposed in *Ctcf-Wapl* DKO cells[37].

Although the precise mechanism of how cohesin islands are triggered during senescence is unclear, it is tempting to speculate that the initial de novo cohesin accumulation promotes new loop formation, and thus, increased gene expression. This would further promote transcription-dependent cohesin accumulation, constituting a gene amplification feed-forward mechanism and eventually contributing to forming the phenotype specific gene expression profile. The enrichment of active enhancer marks on cohesin islands in RIS cells might also contribute to the ability of cohesin islands to form new interactions (Supplementary Fig. 10f). Our data highlight that such accumulation of cohesin islands does occur in physiological contexts in mammalian cells, where they potentially constitute an additional layer of gene regulation for cell fate determination, by modulating higher-order chromatin structure.

## Methods

**Cell culture.** IMR90 and WI38 HDFs (ATCC) were cultured in Dulbecco's modified Eagle's medium (DMEM)/10% foetal calf serum (FCS) in a 5% $O_2$/5% $CO_2$ atmosphere. Cell identity was confirmed by STR (short tandem repeats) genotyping. Cells were regularly tested for mycoplasma contamination and always found to be negative. The following compounds were used in cultures: 100 nM 4-hydroxytamoxifen (4OHT) (Sigma, H7904), 100 μM 5,6-dichloro-1-β-D-ribofuranosylbenzimidazole (DRB) (Sigma, D1916), 10 ng/mL tumour necrosis factor alpha (TNFα) (PeproTech, 300-01 A) as indicated in individual figures. TNFα was added to the culture media to the final concentration of 10 ng/mL for 1 h before harvesting for ChIP-seq and immunofluorescence studies.

**Vectors.** The following retroviral vectors were used: pLNCX2 (clontech) for *ER:HRAS^{G12V}* (Young et al.[21]), pBabe-puro for *HRAS^{G12V}*. Senescence was induced using the ER:RAS system unless otherwise mentioned.

**DNA-FISH**. DNA-FISH was performed as previously described[46]. Cells were plated onto glass coverslips the day before fixation. Cells were pre-treated with digitonin (150 μg/ml) in Cytoskeletal (CSK) Buffer (100 mM NaCl, 300 mM sucrose, 3 mM MgCl$_2$, 10 mM PIPES pH 6.8), then fixed in 4% paraformaldehyde. Cells were permeabilized with 0.2% (v/v) Triton X-100 (Sigma, X100) in PBS, soaked in liquid nitrogen, treated in 0.1 M HCl and dehydrated in EtOH. We used the fluorescent labelled probes corresponding to the following BAC clones (Empire Genomics): 5-fluorescein-labelled RP11-57I3 (*NRG1* gene body, 'gene probe'), 5-TAMRA-labelled RP11-451O18 (*NRG1* neighbouring region, 'TAD probe'), 5-fluorescein-labelled RP11-185D13 (*HMGA2* gene body, 'gene probe') and 5-TAMRA-labelled RP11-63F4 (*HMGA2* neighbouring region, 'TAD probe'). Confocal images were obtained using a Leica TCS SP8 microscope.

Analysis of FISH signal was performed using scikit-image ([https://scikit-image.org]), by filtering noise using Yen filters for the image channels with FISH signal and Li filter for the DAPI channel. The Clear Border segmentation algorithm was used both for determining individual cell nuclei from DAPI signal, as well as for detecting areas with high FISH signal. We determined the average distance between the peaks of the two types of FISH signals in each cell and then performed two-sided Student t-tests to determine whether distances between probes are significantly different between conditions.

**Immunofluorescence**. Immunofluorescence staining was performed as described previously[46]. Briefly, cells were plated onto #1.5 glass coverslips coated with 0.1% gelatin (StemCell Technologies, 07903) the day before fixation. Cells were fixed in 1% (v/v) paraformaldehyde in PBS for 1 h at 4 °C and permeabilised with 0.2% (v/v) Triton X-100 (Sigma, X100) in PBS for 5 min at room temperature. The primary antibody used was anti-RELA (Cell Signalling Technology, 8242, lot # 13, 1:400) in PBS + 0.1% Tween 20 (Sigma P1379) (PBS-T). The secondary antibody used was goat anti-rabbit IgG (Alexa Fluor 488, Thermo Fisher, A-11034, lot # 2069632, 1:1000) in PBS-T. Cells were counter-stained with DAPI at 1 μM in the secondary antibody solution. Coverslips were mounted onto Superfrost Plus slides (Thermo Fisher, 10149870) with Vectashield Antifade mounting medium (Vector Laboratories Ltd. H-1000). Fluorescence images were obtained using Leica DMI6000B epifluorescence light microscope.

**ChIP-seq**. Chromatin immunoprecipitation (ChIP) was performed as previously described[47,48] for the following antibodies: anti-H3K27ac (Hiroshi Kimura Laboratory, clone CMA309[49], 10 μg/20 M cells), anti-H3K27me3 (Hiroshi Kimura Laboratory, clone CMA323[49], 5 μg/10 M cells), anti-CTCF (Cell Signaling Technology, 3418, clone D31H12, lot #1, 10 μL/20 M cells), anti-RAD21 (Katsuhiko Shirahige Laboratory[50], 10 μg/20 M cells) and anti-SMC3 (Abcam ab9263, lot # GR290533-17 and GR3221084-8, 10 μg/20 M cells), anti-Pol II (Hiroshi Kimura Laboratory, clone C13B9[49], 5 μg/10 M cells). Libraries were prepared using the NEBNext Ultra II DNA Library Prep Kit for Illumina (New England Biolabs, E7645L) according to the manufacturer's instructions except that size selection was performed after PCR amplification using AMPure XP beads (Beckman Coulter, A63881). Samples were sequenced paired-end using 50 bp reads on the Illumina platforms.

**Quantitative reverse transcription-PCR**. RNA was prepared using the Qiagen RNeasy Plus Kit (74136, Qiagen) according to the manufacturer's instructions and reverse-transcribed to cDNA using the Applied Biosystems High-Capacity Reverse Transcription Kit (43-688-13, Thermo Fisher). Relative expression was calculated as previously described[31] on an Applied Biosystems Quantstudio 6 by the $2^{-\Delta\Delta Ct}$ method using β-actin (*ACTB*) as an internal control. The primers used are mentioned in Supplementary Data 7.

**Hi-C and capture Hi-C**. Hi-C and capture Hi-C libraries were generated as previously described[51–53] using the in-nucleus ligation protocol[54]. For each sample and replicate 50 million IMR90 cells were used. For capture Hi-C, biotinylated 120-mer RNA baits complementary to both ends of each target *HindIII* restriction fragment of interest were designed. Target sequences were valid if they contained no more than two consecutive N's, were within 330 bp of the *HindIII* restriction site and had a GC content ranging between 25 and 65%.

**Hi-C data processing**. Hi-C and cHi-C libraries were quality checked with FastQC and aligned with HiC-Pro[55] against the hg19 genome build. Artefacts were identified and removed using both HiC-Pro and diffHic[28] (R Bioconductor package) and reads were counted into bins at several resolutions (*HindIII* and 5 kb for cHi-C and 10 kb–100 kb for Hi-C). Read duplicates were removed using samtools[56] markdup. We used HiC-Spector[22] and HiCRep[23] to check for the similarity between biological replicates.

**A/B compartments**. A/B compartments were called as before[24], by performing PCA on distance-corrected, ICE-normalized Hi-C matrices at 100 kb resolution. The principal component which correlated well in absolute value with H3K4me1 ChIP-seq signal was chosen as representative of A/B compartments. The sign of the A/B compartments vector was set to match the sign of the correlation with

H3K4me1 signal so that A compartment regions were represented by positive values and B compartment regions were represented by negative values.

**TADs**. TADs were called using TADbit[30] from Hi-C matrices at 40 kb resolution. A confidence score between 1 and 10 was assigned to each TAD border by TADbit. TADs from different biological replicates were combined in a consensus set per condition using TADbit by only considering common TAD borders with scores over 7 (out of 10). Normalized Mutual Information (NMI) was used to compare the growing and the RIS set of TAD borders for agreement, using the R package NMI (CRAN [https://cran.r-project.org]). The exact positions (bin) of the TAD borders were used to compare growing and RIS borders.

**Interaction modelling**. We used TADbit[30] to compute 3D models of the interactions of genes released from H3K27me3 neighbourhoods using the ICE-normalized matrices at 20 kb resolution, combined across biological replicates for growing and RIS, respectively. The matrices used correspond to the subset of interactions of one of two TADs around each gene of interest. In each case, we tried several parameter spaces for IMP parameter optimisation, employed by TADbit. For each region, we then chose the parameter subspace which fit the interaction values curve best. Modelling with IMP[57] within TADbit was then performed with the parameters optimised for each case. The top 10 models predicted in each case were selected and exported from TADbit as XYZ coordinates.

**Differential interaction analysis**. We performed differential interaction analysis between growing and RIS Hi-C and cHi-C libraries at several resolutions (*HindIII* and 5 kb for cHi-C and 10 kb–100 kb for Hi-C, increasing in 5 kb steps) using diffHic[28] (R Bioconductor). Libraries with artefacts and duplicates removed were further filtered for low counts and diagonal entries. Using diffHic, we performed non-linear normalization (LOESS) to remove trended biases between libraries. We tested for significant interaction changes at 5% FDR by using quasi-likelihood F-tests and Benjamini–Hochberg multiple testing correction from diffHic.

Enhancer-promoter interactions were annotated by checking the bins involved in significant differential interactions for overlaps with enhancers and promoters. We used cHi-C EP interactions annotated using *HindIII* fragments and we combined Hi-C EP interactions determined at 10 kb, 15 kb, 20 kb, 25 kb, and 30 kb, filtered for enhancers longer than 7.5 kb. Enhancers were determined as before[6], using H3K27ac peaks which overlap ATAC-seq peaks in each condition and collapsing peaks nearer than 12.5 kb. Promoters were represented as 5 kb regions around the TSS of protein-coding genes (GENCODE v19 reference). Only promoters of differentially expressed genes in RIS were considered.

**Using cHi-C to filter EP interactions determined with Hi-C**. In order to annotate EP interactions from Hi-C more robustly, we compared several filtering strategies at different resolutions, using the contacts detected using cHi-C as a baseline for comparison, due to their accuracy at high resolution (*HindIII*). We wanted to maximise the number of EP interactions detected in the captured regions from both Hi-C and cHi-C and to minimise the interactions detected from Hi-C but not from cHi-C, which were likely false positives. We tried selecting only interactions involving enhancers of large sizes (over 5 kb, 7.5 kb or 10 kb) or genes which were more robustly differentially expressed in RIS (FDR < 0.01), as well as selecting bins without other regulatory elements. All of these filters were applied on EP interactions detected at resolutions between 10 kb and 100 kb as bin size can also affect the accuracy of the interactions detected. Finally, we selected resolutions higher than 30 kb (10 kb, 15 kb, 20 kb, 25 kb, 30 kb) and interactions involving enhancers larger than 7.5 kb for the EP interactions annotated from Hi-C.

**ChIP-seq analysis**. ChIP-seq libraries were aligned against the hg19 genome build using bowtie2[58] and uniquely mapping reads which did not bind to 'blacklisted regions'[43,59] were used for further analysis. Peak calling was performed using MACS2[60] with insert sizes calculated using the R Bioconductor ChIPQC package[61]. Consensus peak sets were calculated for each condition by selecting peaks which appear in at least two replicates. Differential binding analysis was performed using THOR[62] and genomic regions were filtered for significant binding changes at 5% FDR and a minimum of 100 reads per location in at least one of the conditions. DeepTools[63] was used to calculate and visualize ChIP-seq profiles summarized across genomic regions.

**RNA-seq analysis**. RNA-seq libraries were aligned using STAR[64] against the hg19 genome build and reads were counted against genes (GENCODE v19 reference [https://www.gencodegenes.org/human/release_19.html]) using subread[65]. Differential expression analysis was performed using glmTREAT from edgeR[66] and significantly differentially expressed genes were selected at 5% FDR. log-TPM expression values were also calculated for the analysis of transcription and cohesin binding. Gene enrichment analysis of sets of genes of interest was performed using the enrichR R package (CRAN [https://cran.r-project.org]) which queries EnrichR[67,68] against the WikiPathways database.

**Data visualization**. Hi-C matrices corresponding to combined biological replicates were used for visualization. The raw matrices for each replicate were combined by calculating the overall negative binomial mean contacts with correctedContact from diffHic[28] and further normalized using ICE[24] and distance correction as part of the correctedContact functionality. Plotting the matrices was performed using a custom set of R scripts consisting of horizontal rotation of the matrix coordinates and overlaid genomic information such as enhancer and promoter positions, ChIP-seq tracks or interaction arcs. Interactions were coloured using a non-linear scale which represented interaction values above and below the expected values (positive and negative values, respectively, determined with distance correction) with 'warm' and 'cold' colours, respectively. In the case of overlaying cHi-C matrices at HindIII resolution, each interaction unit was represented as equally sized, despite the variable lengths of HindIII fragments. The R scripts used for this visualisation strategy were made available as a package called hicvizR (https://gitlab.com/ilyco/hicvizr). ChIP-seq tracks represented were THOR-normalized (input subtracted and library-normalized) bigWig files produced during the differential binding analysis. The tracks were exported from IGV[69] and were scaled to be within the same interval, to allow for comparison between conditions. RNA-seq bigWig files were also produced for visualization of expression and TMM factors were used for normalization of the signal between libraries. Models derived from Hi-C interactions using TADbit[30] were visualized using the R package rgl (CRAN [https://cran.r-project.org]). The points corresponding to the model are centred around 0. The curve used to visualize the model is drawn by adding 10 additional points between every pair of points in the set of original coordinates by interpolation with the spline function in R. The radial image of loops with cohesin/CTCF changes in Fig. 3b was achieved using ggplot2 R package by representing each loop segment (genomic region between the two loop ends) radially and scaled so that all the segments have the same length. Different chromosomes are segregated by white space and CTCF/cohesin changes are plotted as dots whose position is relative to the loop segment.

**Cohesin islands**. We investigated the association between cohesin accumulation and transcription by grouping genes by expression level (represented as log-TPM averaged across biological replicates) and plotting their SMC3 and RAD21 ChIP-seq profile. The binding profile was centred either at their transcription end site (TES) in the case of isolated genes or in the middle of the genomic region bounded by TES of two convergent genes. We focused this analysis on isolated and convergent genes, like in Busslinger et al.[37], in order to avoid biases caused by genes with overlapping regions post 3′UTR. Small increases in signal were observed before the TES as well as caused by short genes which show cohesin binding on their gene bodies as well.

We determined cohesin islands by comparing the RIS cohesin ChIP-seq libraries with and without DRB treatment (transcription elongation inhibitor). We then selected significantly differentially bound regions larger than 2 kb. We eliminated possible false positives which can occur due to genes with overlapping 3′ end regions (such as convergent genes) by filtering for highly expressed genes. Genes with cohesin islands were determined by overlapping 10 kb regions starting at the TES with the cohesin islands determined.

**Interaction neighbourhood aggregation**. We identified general trends of certain subsets of interactions by selecting a two-dimensional neighbourhood around each interaction of interest, and summing the corresponding Hi-C sub-matrix (from the ICE-normalized and distance-corrected matrix averaged across replicates, as described earlier), similarly to Aggregate Peak Analysis[13]. Each interaction pixel was divided by the number of sub-matrices added minus the number of missing values. We selected a 200 kb region around each bin containing a cohesin peak of interest which, at 20 kb resolution, resulted in neighbourhoods of $11 \times 11$ pixels. Differential aggregated matrices were computed by subtracting the growing-specific aggregated matrix from the RIS one.

**Reporting summary**. Further information on research design is available in the Nature Research Reporting Summary linked to this article.

## Data availability

Hi-C and cHi-C data in growing and senescent IMR90 cells, as well as ChIP-seq data in IMR90 and WI38 human diploid fibroblasts in the growing (with and without TNFα treatment) and RIS (with and without DRB treatment) conditions were deposited in the Gene Expression Omnibus: GSE135093. Publicly available data in growing and senescent IMR90 cells were reanalysed from our previous studies: H3K4me3 and H3K27me3 ChIP-seq from Chandra et al.[25] ("GSE38448"), H3K27ac ChIP-seq and ATAC-seq from Parry et al.[27] ("GSE103590"), RNA-seq data from Hoare et al.[26] ("GSE72404"). The following external datasets were reanalysed: RAD21 and CTCF ChIP-seq in monocyte (THP-1) and macrophage (PMA-induced) controls from Heinz et al.[45] ("GSE103477"), RNA-seq and Hi-C in monocyte (THP-1) and macrophage (PMA-induced) controls from Phanstiel et al.[44] ("GSE96800", "PRJNA385337"), RNA-seq and Hi-C in IMR90 cells with or without TNFα treatment from Jin et al.[15] ("GSE43070"), Hi-C data in growing and senescence WI38 cells from Chandra et al.[7] ("PRJEB8073"). All other relevant data

supporting the key findings of this study are available within the article and its Supplementary Information files or from the corresponding author upon reasonable request. Source data for Fig. 1e, Supplementary Fig. 2c and 7b are provided with this paper (Supplementary Data 8). A reporting summary for this Article is available as a Supplementary Information file.

## Code availability

Custom scripts used for enhancer-promoter annotation and filtering THOR differential binding output were uploaded to the OSF public repository ([https://osf.io/xajd3/?view_only=6860fe4b8421475485b7e251d735db58]). A package for visualization of Hi-C matrices is also available on GitLab ([https://gitlab.com/ilyco/hicvizr]).

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

## Acknowledgements

We thank all members of the Narita laboratory for helpful discussions, CRUK-CI core facilities (Genomics, Biorepository, Bioinformatics and Research Instrumentation) for technical support. This work was supported by a Cancer Research UK Cambridge Institute Core Grant (C9545/A29580) to the M.N. laboratory. M.N. was also supported by the Medical Research Council (MR/M013049/1) and Biotechnology and Biological Science Research Council (BB/S013466/1). I.O. was supported by Wellcome Trust (105367/Z/14/Z). S.S. was supported by the UKRI Biotechnology and Biological Science Research Council (BB/J004480/1), the UKRI Medical Research Council (MR/T016787/1), and a Career Progression Fellowship from the Babraham Institute. A.J.P. was supported by a Sir Henry Wellcome Postdoctoral Fellowship (215912/Z/19/Z). H.K. was supported by JSPS KAKENHI (JP17H01417 and JP18H05527) and JST-CREST (JPMJCR16G1). K.S. and M.B. were supported by Grant-in-Aid for Scientific Research on Innovative Areas (15H05976). S.A.S and D.B. are supported by the Medical Research Council (MC_UU_12022/10). P.F. was supported by the UKRI Medical Research Council UK (MR/L007150/1) and the UKRI Biotechnology and Biological Science Research Council UK (BB/J004480/1).

## Author contributions

I.O. and M.N. conceived the study. A.J.P., S.S. and YI performed Hi-C and capture Hi-C experiments. P.F. supervised Hi-C/capture Hi-C experiments and interpreted the data. Masako N. and A.J.P. performed the ChIP-seq experiments with help by H.K., K.S. and M.B. Masako N. performed the DNA-FISH experiments. A.S.L.C performed the TNFα experiments. G.S. performed three-dimensional interaction modelling and visualization. I.O. analysed the Hi-C, cHi-C, ChIP-seq and RNA-seq data with input from D.B., S.A.S. and A.S.L.C. M.N. and I.O. wrote the paper with input from the other authors.

## Competing interests

P.F. and S.S. are co-founders Enhanc3D Genomics Ltd. All other authors declare no competing interests.
