## [Peer Review File · Nature Communications]

Reviewers' comments:

Reviewer #1 (Remarks to the Author):

Overall comment

The authors have demonstrated that there are EP changes, cohesin loops and island formation, and transcriptional changes upon RIS. While the manuscript nicely describes many of these changes in interaction and transcription, the majority of the relations between EPs, cohesin and transcription, are co-occurrence correlation in nature. Overall, the manuscript lacks a genome-wide description of the events that they observe and describe for individual genes, so it remains unclear whether these are cherry-picked examples (and thus potentially artifacts of the data), or indeed constitute a general trend. Further, the causality of the events as claimed and concluded in many paragraphs have not been demonstrated and should either be addressed by more experimentations at the specific locus proposed (namely IL1 and/or MMP) based on genome-wide statistical analyses. Overall, the manuscript uses genome-wide data to infer mechanisms and interactions at individual loci and genomic regions, without any validation. This is very dangerous since it is highly likely to pick up outliers and false positives and doesn't put the results into context in terms of what is expected under the null hypothesis of no relationship.

Major comments:

- 1) A global description of the dataset is completely lacking. The figures describe all individual loci from which the authors make conclusions. Ideally, they would use these snapshots to identify pattern and then validate them globally using statistical analysis across the whole genome.
- 2) The authors claim good agreement of biological replicates in HiCSpector and PCA (extended figure 1a - c), since on the PCA there was quite a big disagreement between the RIS replicates. Further it was unclear in what sense the replicates are in good agreement of (for instance, read count / reproducibility of the TAD domains).
- 3) The authors picked the NRG1 locus as one of the most upregulated locus in RIS (figure 1d) without giving a global overview of other loci. To arrive at this conclusion from genome-wide data the authors should do a proper statistical analysis of the RNA-Seq, instead of just showing the track (e.g. using DeSeq2 or edgeR). Further, it was unclear how this statement was related to the HiC map (figure 1d), which showed a down-regulation of chromatin contact.
- 4) Similarly, the authors described an increased ATAC-Seq signal across the NRG1 gene in the RIS condition. However, the ATAC-Seq track signal did not show the described "increase in chromatin accessibility", and the authors did not perform statistical analysis to underline their claim (using DiffBind for example). This has to be done.
- 5) The authors wrote that the properties of the HMGA2 gene (extended figure 1d) between the RIS and Growing conditions are similar to that observed in the NRG1 gene. While the HiC contacts are indeed reduced in the RIS condition, there is almost no visible difference between the gene expression signal on the RNA-seq track. Similar to the above, a relevant statistical test has to be performed for such claims.
- 6) for the statement on line 93: "[we] identified 102 up-regulated genes dissociating from H3K27me3 regions" it is very unclear where it comes from, and what the underlying data / statistical analyses are that lead to the statement. This would also be a prime example where the authors could take some of the observation to a genome-wide level by assessing whether genome-

wide H3K27me3 is associated with reduced gene expression overall.

7) for the statement on Line 94 - 97: "These data suggest that H3K27me3 regions might contribute to long-range silencing of neighbouring genes through 3D positioning within TADs and that release from such domains appears to be a relatively common mechanism of gene activation during RIS." it is unclear from which data this conclusion is drawn. First, it is unclear what "these data" is. Second, the authors did not explore any "long-range" gene silencing at all by H3K27me3, for example illustrating "long-range" interactions between H3K27me3 regions and silenced regions within a TAD. third it is actually unclear how long (in terms of kb) "long-range" is here. Finally, there is no study about the relation between H3K27me3 and the HiC contacts, RNA-seq ATAC-seq signal, and the distances of the interactions at all in the whole figure. Further analyses will be required here to explore and define clearly the relation between H3K27me3 and the rest of the data for this conclusion to be valid.

8) The authors wrote "genes differentially expressed during RIS[27]", it is unclear from the text and the extended figure 3 & 4a whether the genes differentially expressed came from this study or from the cited study numbered 27 (Hoare et al. 2016 Nat Cell Biology). If indeed the differential genes are from the cited study, the authors should re-do the analysis using their own data, so as to correctly match the HiC and capture HiC data.

9) For the statement "The complex rewiring was exemplified by the IL1 and MMP loci...", it is not shown in the figure 2a and extended figure 4a, how the IL1 and MMP loci being "rewired" has what effects on the genes. For making this claim, the global relations between this "rewiring" should be analysed, also especially with the given specific examples. For instance, an alternative interpretation of the MMP10 example (extended figure 4a) associates with 5 decreased interactions with enhancers, and 3 increased interactions with enhancers, while gene expression is up-regulated. One could hypothesise that the 3 enhancers are stronger enhancers than the 5 combined. MMP1 loses interactions with 2 out of the 3 "stronger" enhancers (extended figure 4a), and yet the expression still increases.

While there is no doubt that a lot of the enhancers are "interacting" with the different genes, it is very unclear how many of these interactions or rewiring are actually significant or important for changing gene expressions. To make any claims about examples, the authors should put this into context of a global analysis.

Also, the authors did not exclude increased gene expression due to for example increased transcription start site chromatin accessibility or activation histone mark for instance.

To really establish the link between EP changes and gene expression further analyses and/or experimentation on specific loci, are required to characterise and disentangle the mechanisms of this "complex rewiring" as the authors have put it.

10) Based on gene ontology enrichment of up-regulated genes in inflammatory terms and down-regulated genes in cell cycle terms, the authors concluded "This suggests that the two senescence hallmarks, the SASP and proliferative arrest, are controlled through the rewiring of the EP network.". However, as pointed above, the actual relation between the rewiring of enhancer promoter pairs remain unexplored in this study, and therefore this conclusion cannot be drawn. The results could only allow the authors to conclude that gene expression changes in the RIS condition indeed agrees with the senescence phenotype as shown in GO enrichments. Further experimentation on specific loci using methods such as CRISPR + capture HiC + qPCR should be required to establish a clear mechanism between the EP and gene expression changes.

11) Based on the co-occurrence of cohesin binding and EP contacts at the same site at the IL1 locus, the authors concluded that "Collectively, these data suggest that the de novo cohesin peak contributes to the formation of new loops in the IL1 locus and that within each loop domain, EP pairs might preferentially contact (Fig. 3g)."

However, the data only showed the co-occurrence of EPs and cohesin loops, and the authors did not demonstrate that cohesin indeed contributes to the formation of the loops at the IL1 locus. Experiments to demonstrate this causal relation should be performed in order for this conclusion to be made.

12) Based on the results in figure 4 and extended figure 9, the authors concluded that "cohesin-mediated loop alteration at the IL1 locus (as in RIS) facilitate transcription of genes in this locus". However, it would appear that, based on their experiments using DRB, transcription would facilitate the formation of cohesin loops. The causal effect between transcription and cohesin loops at this particular IL1 locus has not been demonstrated and should be determined experimentally. For instance, by removing the cohesin binding sites using CRISPR and determine how gene expression is affected. Or globally by overexpressing cohesin to determine if transcription is affected in order to establish the mechanisms.

Line 279

As previously mentioned, the authors did not show an association between altered EP and gene expression, and should be determined by further experimentation as mentioned above.

Minor comments:

a) The correlation between SMC3 and RAD21 genome wide chip seq signal should be shown in supplementary, including p-values.

b) we are unable to find the relevant section in the methods that is supposed to explain figure 3b regarding CTCF and cohesin binding changes.

c) The authors should show an example of the data of where the cohesin "anchor" is in relation to the CTCF in the extended figure. Also describe the parameters of how cohesin anchor is defined.

d)The method description is missing related to figure 3c, regarding CTCF and cohesin loops

e) Figure 1: The authors did not clarify whether the illustrated figures are representative of both replicates.

Reviewer #2 (Remarks to the Author):

The authors performed a study of the 3D rewiring of chromosomes in oncogenic HRAS-induced senescence by combining Hi-C and capture Hi-C, and associate it with cohesin redistribution. This is an important and topical study shedding new light on the chromosome conformation changes driving the expression of SASP genes in oncogene induced senescence. However, some of the results presented should be better clarified and validation is partially missing.

Major Revisions:

1. The authors should better clarify how enhancer regions are defined. Were histone marks or

enhancer RNA's considered? If so, please emphasize this data more. It is mentioned that EP were considered only if enhancers were between 7.5 KB and smaller than 30 KB. However, this does not address how enhancer are defined in this study.

2. Are the EP pairs found in senescence due to the formation of novel enhancers? Please discuss this more in the main body of text.

3. Line 279: It would be informative to do a motif analysis for TFs in loops/enhancers that forms in response to TNF-alpha treatment compared to those that form in senescence.

4. No validation is presented for any of the interactions discussed in the manuscript. For example, the dissociation of NRG1 from the local chromatin environment or the loop formation of IL1B could be verified with microscopy.

Minor Revisions:

1. Please specify the resolution for the contact matrices when comparing biological replicates and calling TADs in the main body of text.

2. Line 80: TAD borders are compared between conditions. What window size is used to compare TAD borders? Please specify in figures and in main body of text.

3. Line 175: Why would CTCF binding increase in senescence if cohesin binding is decreased? Are there any other known roles of CTCF binding that are independent of cohesin formation?

4. Line 195: This sentence is confusing, please clarify. Are you saying that the regions where CTCF decreased in binding in RIS did not result in complete loss of CTCF at these sites?

5. Line 408: HiC protocols generally call for 2-5 million cells per replicate. Is '50 million' a typo?

6. Line 424: How strong is the correlation between H3K4me1 signal and the principal component? Please show data, perhaps a meta plot.

7. Line 463: The rigor here is appreciated, but why would these interactions be "false positives"? Is this due to the limited resolution in Hi-C libraries as compared to capture Hi-C libraries?

8. Please discuss further why cohesin islands are more present in RIS conditions when compared to proliferating controls.

9. Methods-Cell Culture: Please give details on TNF alpha treatment. What marker genes were tested to verify the efficacy of the treatment. How long did the treatment last? Was FBS concentration reduced? Is there TNF-alpha in the FBS already?

Reviewer #3 (Remarks to the Author):

In this work, the authors examined the changes in chromatin structure that are associated with the changes in expression that occur during oncogene-induced senescence in human diploid fibroblasts. Using in situ Hi-C, together with previously obtained genome-wide data, they first identify 102 up-regulated genes that dissociate from H3K27me3-enriched regions and suggest that this may be a "relatively common mechanism of gene activation" during this senescence (p. 4). They then combined in situ Hi-C with capture Hi-C, and previously obtained data, to identify 1004 enhancer-promoter (EP) interactions that change significantly during this senescence and that are associated with differential expression. The genes associated with these altered EP interactions were found to be enriched for "inflammatory" and "cell cycle arrest" genes, which led the authors to conclude that the two "hallmark" features of senescence, the senescence-associated secretory phenotype (SASP) and proliferative arrest, "are controlled through the rewiring of the EP network" (p. 5). Finally, as the proteins CTCF and cohesin are believed to be directly responsible for the formation of chromatin loops that underlie many EP interactions, the authors used ChIP-seq to characterize their genomic location and extent of binding during senescence. They found changes especially in cohesin binding that are associated with alterations in many loops. Yet perhaps most intriguingly, they identified 574 extended regions of de novo cohesin binding that resemble previously observed "cohesin islands" in (CTCF and Wapl) double knockout (DKO) mouse embryonic fibroblasts. By resemblance, the authors note an enrichment at the 3' end of expressed genes, an elongated shape, as well as a loss of binding upon transcription inhibition, but could have (probably) also mentioned a more extended binding region than is typically observed

associated with CTCF. They also note similar findings upon re-analysis of previously published monocyte-to-macrophage data. With this, the authors conclude that these cohesin islands do indeed “occur in physiological contexts in mammalian cells” (p. 12), where they could play a role in gene regulation during cell fate determination.

Overall, this is a very good piece of work: it is well written, thoroughly analyzed, and contains a copious amount of data that largely supports their main conclusions. At the moment, there is considerable interest in understanding the biological processes underlying senescence, as well the link between chromatin structure and gene expression. This work is a useful, and perhaps significant, contribution in both areas. My prime concern is the absence of analysis that quantitatively supports the conclusions that these identified structural changes are prominent mechanisms by which the senescent phenotype is effectuated.

Specifically, my major concern is:

1. The authors should mention the number of genes, overall, that are differentially expressed, as well as up- or down-regulated. It is otherwise difficult to judge if indeed the 102 that dissociate from the H3K27me3-enriched regions do reflect evidence for a “common” mechanism of gene regulation in senescence. Similarly, a quantification of the total number of differential expressed genes that are associated with the two senescence hallmarks is also necessary to determine if the fraction associated with changed EP interactions is significant enough to justly render the re-wiring of this network as “controlling” these two hallmarks.

Other concerns are:

1. While some of the observations here do resemble the “cohesin islands” previously observed in DKO cells, there are also significant differences. In particular, those observed here are found to be associated with genes that are specifically enriched for pathways important in senescence, while no such selectivity (based on gene function) is apparently present in the DKO cells. In addition, while those in the DKO study are found to redistribute to the transcription start-site (TSS) upon transcription inhibition, those observed here do not (Fig 4b). Those observed here also seem to extend across the entire gene, in contrast to those in DKO cells. The local minimum specifically at the TES in Fig 4a and Extended Data Fig 9b is also unlike the corresponding profile from the DKO study. Also, one might predict that CTCF binding within the MMP1 gene would prevent cohesin island formation (according to the model presented in the DKO study), but it clearly does not here (MMP1 in Extended Data Fig 8). Finally, all of the cohesin islands in the DKO study were located downstream of actively transcribed genes, whereas with 574 islands overlapping near the 3’ ends of 343 genes in this study (p. 10), I believe that would leave 231 islands that are not associated with the 3’ ends of genes. It is thus not clear that the mechanism underlying their formation is the same as in the DKO cells. As such, there is no evidence presented here to justify the phrase, “transcription-driven” in the title. In the DKO cells, it was the repositioning to the TSS upon transcription inhibition and the changes in peak shape between converging genes as a function of the relative extent of expression that were consistent with a “transcription-driven” mechanism. Perhaps “dependent” instead of “driven” would be more appropriate in the title. In addition, the authors may wish to include some discussion of these differences with the DKO cells, as they may help future efforts to uncover the mechanism.

2. The PCA results in Extended Data Figure 1 indeed show that the “Growing” replicates cluster together, but strangely, the RIS replicates do not. It is not clear that one could conclude from this calculation that the RIS replicates are indeed consistent with each other, although the results from HiCSpector look good. Perhaps the authors could provide some additional discussion for the PCA results or use another tool, such as HicRep, to further substantiate agreement between the RIS replicates.

3. Were the EP interactions from the Hi-C data determined from only one replicate? If so, how do

they compare with those determined from the other replicate?

4. By the authors' own criteria of filtering the Hi-C EP interactions, I believe that Extended Data Fig 4b shows that enhancers larger than 5 kb would have been a better choice than those larger than 7.5 kb.

5. On p. 6, the authors describe the formation and loss of many loops within the IL1 cluster based on their cHi-C data (Fig 2f), while on p. 8, line 197, the authors mention that "The genes and enhancers studied in the IL1 locus belong to a single loop identified in IMR90 cells (ref 8). This is consistent with enhancer sharing between these genes...". How could the presence of a single loop in IMR90 be consistent with enhancer sharing that the authors articulate occurs with the formation and loss of the many loops that are shown in Fig 2f?

6. I am a little confused by the following numbers: the author state that "3407 (out of 7,647) loops showed changes in cohesin binding, mostly decreases (80%)" (line 180, p. 7), which corresponds to 2726 loops. But later (line 187), the authors mention that "Loops with cohesin loss at one or both ends (1,827 loops)...". What is the reason for the difference of nearly 900 loops?

7. With respect to Fig 3g, the new cohesin binding site at IL1B was shown to interact more with the anchors of the loops (Fig 2d,f). That is, with the anchors to the right in Fig 2f. There are no anchors to the left in this image and there is no evidence for increased contact between the cohesin binding site in the Hi-C with any putative anchors on the left. Yet, I believe Fig 3g depicts contact between the cohesin site and an anchor on the left.

8. I did not understand what the authors meant by (line 189, p. 8) "More stringently, 326 loops (Fig. 3c, right) were found to overlap with significantly reduced interactions during RIS, either from cHi-C or Hi-C (at 20 kb and 40 kb resolution)". What does "more stringently" mean here? The figure legend indicates "Compare to all IMR90 loops with significantly decreased interactions during RIS (right)", which did not help.

9. In line 228, p. 9, the authors refer to Fig 4a as evidence for the accumulation of cohesin islands and cohesin binding that is correlated with gene expression, but this figure only shows an increase in cohesin binding, not the accumulation of cohesin islands. However, I believe that showing such an enrichment should be possible with their data.

10. Did transcription inhibition only reduce the binding at cohesin islands or also at "typical" cohesin binding sites (that is, those associated with CTCF)? Providing this information might also help future efforts aimed at resolving the mechanism.

11. A few concerns with the figures: in Fig 2f, the promoters for all genes, especially CKAP2L, are at the 3' ends of the genes; in Fig 3a, the y-axis is number, not frequency; in Extended Data Fig 1d, the blocks labeled HMGA2, IRAK3, and GRIP1 in the Hi-C map do not align with the genes in the track at the bottom of the figure; and the legend to Extended Data Figure 3b appears to be missing the words "active promoters" associated with the low/high histone modifications.

Reviewers' comments:

Reviewer #1:

Overall comment

The authors have demonstrated that there are EP changes, cohesin loops and island formation, and transcriptional changes upon RIS. While the manuscript nicely describes many of these changes in interaction and transcription, the majority of the relations between EPs, cohesin and transcription, are co-occurrence correlation in nature. Overall, the manuscript lacks a genome-wide description of the events that they observe and describe for individual genes, so it remains unclear whether these are cherry-picked examples (and thus potentially artifacts of the data), or indeed constitute a general trend. Further, the causality of the events as claimed and concluded in many paragraphs have not been demonstrated and should either be addressed by more experimentations at the specific locus proposed (namely IL1 and/or MMP) based on genome-wide statistical analyses. Overall, the manuscript uses genome-wide data to infer mechanisms and interactions at individual loci and genomic regions, without any validation. This is very dangerous since it is highly likely to pick up outliers and false positives and doesn't put the results into context in terms of what is expected under the null hypothesis of no relationship.

Major comments:

1) A global description of the dataset is completely lacking. The figures describe all individual loci from which the authors make conclusions. Ideally, they would use these snapshots to identify pattern and then validate them globally using statistical analysis across the whole genome.

The global description of the dataset in terms of sample description (nr. reads, agreement), as well as TADs distribution, A/B compartments and number of significant interaction changes, was briefly described in lines 78-81 on page 3 of the original version (now lines 122-126) as well as Fig. 1b and Extended Fig. 2 (now Supplementary Fig. 1 c, e). We would like to clarify that we did exactly what the reviewer suggests: the differential interaction analysis between the two conditions was performed genome-wide using Hi-C data, as well as for the chosen loci in the capture Hi-C dataset. To further emphasize this important point, we have now provided additional information and figures regarding the TAD ranking by most interaction changes and the top TADs affected by changes during senescence which include *NRG1* and *HMGGA2* TADs (new Supplementary Fig. 1f, g). We also updated Supplementary Fig. 1e to correspond to all significant interaction changes at resolutions between 10 and 100kb to give a better global overview of all the changes occurring in RIS from which we later focus on subsets of interest such as intra-TAD interactions, genes-H3K27me3 or genes-enhancers interactions.

2) The authors claim good agreement of biological replicates in HiCSpector and PCA (extended figure 1a - c), since on the PCA there was quite a big disagreement between the RIS replicates. Further it was unclear in what sense the replicates are in good agreement of (for instance, read count / reproducibility of the TAD domains).

HiCSpector agreement scores reflect ‘good agreement’ between interaction read counts (above 0.75 out of a score between 0 and 1, where 1 indicates good agreement). We used PCA to show that the main effect studied is indeed obvious in the first principal component (Growing vs RIS) and that the conditions are separable. As for the agreement between samples, the two RIS samples are not necessarily in disagreement because in PC3 and PC4 they are clustered together (Fig. R1).

Fig. R1: The relative position of growing and RIS replicates in terms of PC3 vs PC4.

As suggested by reviewer 3, to further increase the confidence of the agreement between biological replicates, we performed HiCRep by Yang et al. 2017 (see image below), which showed high agreement scores between replicates.

Fig. R2: HiCRep reproducibility scores of each pairwise comparison between biological Hi-C replicates, between 0 (poor agreement) and 1 (good agreement) (new Supple Fig. 1b).

We have now added the information about agreement between replicates calculated using both HiCSpector as well as HiCRep in the manuscript (new lines 119-120) and in new Supplementary Figure 1b.

3) The authors picked the *NRG1* locus as one of the most upregulated locus in RIS (figure 1d) without giving a global overview of other loci. To arrive at this conclusion from genome-wide data the authors should do a proper statistical analysis of the RNA-Seq, instead of just showing the track (e.g. using DeSeq2 or edgeR). Further, it was unclear how this statement was related to the HiC map (figure 1d), which showed a down-regulation of chromatin contact.

We would like to clarify that we chose *NRG1* as the locus with the most striking interaction changes based on the genome-wide interaction analysis (using diffHic, which identifies statistically significant changes), as well as up-regulation of the gene in RIS compared to Growing. This was not derived from visual inspection, but from this ranking of most altered TADs.

To further clarify this point, we have provided two new figures showing the ranking of all TADs by their number of differential interactions and the top 10 TADs disrupted (Supplementary Fig. 1f, g), as well as an additional table (spreadsheet Supplementary Table S3) with the coordinates of the top 10 altered TADs.

We performed differential expression analysis on the RNA-seq data using edgeR as mentioned in Methods lines 479-486 (original version, now lines 599-606). All the genes mentioned in the text are significantly differentially expressed as tested with edgeR. In order to clarify this in the text, we have added p-values next to the *NRG1* description (FDR 1.83e-315, logFC 4.56) and other genes mentioned as differentially expressed.

The *NRG1* gene body coincided with the down-regulation of chromatin contacts which are enriched for H3K27me3. The dissociation of *NRG1* from H3K27me3 TAD was correlated with increased ATAC-seq and expression. We have now experimentally validated the structural alteration by DNA-FISH and added this information in the manuscript, as well as in new Figure 1d, e (see also our response to reviewer 2 - point 4).

4) Similarly, the authors described an increased ATAC-Seq signal across the *NRG1* gene in the RIS condition. However, the ATAC-Seq track signal did not show the described “increase in chromatin accessibility”, and the authors did not perform statistical analysis to underline their claim (using DiffBind for example). This has to be done.

We did indeed perform statistical analysis of the differential binding in the original analysis. We used THOR (Allhoff et al. 2016) for all ChIP-seq and ATAC-seq datasets as outlined in the Methods (line 475-477 in the original version; now lines 594-596).

The increase in chromatin accessibility between the Growing and RIS ATAC-seq samples was statistically significant at the *NRG1* locus as shown below (magnified view of the orange

rectangular region in ATAC-seq) with the changing regions as a bed track called THOR differences (filtered by false discovery rate 0.05 and minimum 100 reads per changing region). Green bars correspond to significant increases in signal, whereas red ones correspond to significant decreases in signal. The tracks in original Figure 1d are attached here for comparison with the ATAC-seq increase highlighted. In the figures below, we include both analyses.

We have made this clearer in the text (now lines 144-146) and in the figure legend of new Figure 1b (original Figure 1d), indicating that differential signal analysis was performed with THOR and that the changes at this locus were significant at FDR 0.05.

Fig. R3: Left: ATAC-seq signal in growing and RIS (normalized with THOR and input-subtracted), as well as significant binding changes calculated with THOR at FDR 0.05 over the *NRG1* gene; Right: Highlighted ATAC-seq binding which is expanded in the left image relative to the *NRG1* gene (original Figure 1d, now Figure 1b).

5) The authors wrote that the properties of the *HMG2A* gene (extended figure 1d) between the RIS and Growing conditions are similar to that observed in the *NRG1* gene. While the HiC contacts are indeed reduced in the RIS condition, there is almost no visible difference between the gene expression signal on the RNA-seq track. Similar to the above, a relevant statistical test has to be performed for such claims.

As mentioned earlier, we have performed differential expression analysis with edgeR. Indeed, the upregulation of *HMG2A* in the dataset used in this paper was highly significant with FDR $2.275e-08$ and a log-fold change of 0.7 (fold change 1.6). This information is now added to the text (line 150-152). The significance testing was performed using glmTreat in edgeR which

is a stringent method, considering a minimum log-fold change of interest and calculating the FDR threshold accordingly.

HMGA2 upregulation during senescence is well established (e.g. Narita et al, Cell 2006; Funayama et al, JCB 2006) and here we share our qPCR data of *HMGA2* alteration during RAS-induced senescence (RIS) in new IMR90 samples (which we added in new Supplementary Fig. 2c).

Fig. R4: Up-regulation of the *HMGA2* gene during RIS tested with qPCR.

6) for the statement on line 93: "[we] identified 102 up-regulated genes dissociating from H3K27me3 regions" it is very unclear where it comes from, and what the underlying data / statistical analyses are that lead to the statement. This would also be a prime example where the authors could take some of the observation to a genome-wide level by assessing whether genome-wide H3K27me3 is associated with reduced gene expression overall.

As mentioned earlier, all analyses stem from our initial genome-wide analysis. We searched for genes with a similar pattern to *NRG1*, i.e. genes dissociating from H3K27me3 were selected as follows: the significant interaction decreases were identified genome-wide during RIS using diffHic, then these contact pairs were filtered for interactions where one of the interacting regions overlaps a gene and the other is bound by H3K27me3. We have clarified in the main text this sequence of analysis steps (new lines 152-155).

7) for the statement on Line 94 - 97: "These data suggest that H3K27me3 regions might contribute to long-range silencing of neighbouring genes through 3D positioning within TADs and that release from such domains appears to be a relatively common mechanism of gene activation during RIS." it is unclear from which data this conclusion is drawn. First, it is unclear what "these data" is. Second, the authors did not explore any "long-range" gene silencing at all by H3K27me3, for example illustrating "long-range" interactions between H3K27me3 regions and silenced regions within a TAD. third it is actually unclear how long (in terms of kb) "long-range" is here. Finally, there is no study about the relation between H3K27me3 and the HiC contacts, RNA-seq ATAC-seq signal, and the distances of the interactions at all in the whole figure. Further analyses will be required here to explore and define clearly the relation between H3K27me3 and the rest of the data for this conclusion to be valid.

Regarding ‘these data’, using diffHic, we first found extensive alterations in chromatin contacts during RIS within TADs and between TADs. This is a genome-wide unbiased analysis, not gene-centric analysis. In this result, the top hit of most changing contact-pairs included the *NRG1* body. The *NRG1* gene has numerous isoforms, and the visual inspection of Hi-C and ChIP-seq data exhibited that the major part of the gene was embedded within a H3K27me3-rich TAD in control cells but it appeared dissociated from this TAD.

NRG1 is known to be upregulated during senescence and we also observed its extensive upregulation with increased ATAC-seq signals during RIS in our data. A similar but less pronounced example was *HMGGA2* (a well-known senescence marker). We then focused on genes within the genome-wide differential contact pairs, and found 102 genes that behaved in a similar manner to these two genes (with a statistical significance). In addition to these data in the original version, we have now added DNA-FISH results (new Fig 1d, e) to validate the chromatin structural alteration at *NRG1* and *HMGGA2* loci (see also our response to **Reviewer 2-4**).

We agree that the usage of “long-range” was vague and we have instead clarified that we refer to dissociation of genes from neighbouring H3K27me3 regions, within the same TAD. The median of the distance between genes and dissociated H3K27me3 regions was 321 kb.

Please note, we removed ‘...relatively common mechanism...’ to tone down the conclusion in this part (see our response to **Reviewer 3-1**).

8) The authors wrote “genes differentially expressed during RIS[27]”, it is unclear from the text and the extended figure 3 & 4a whether the genes differentially expressed came from this study or from the cited study numbered 27 (Hoare et al. 2016 Nat Cell Biology). If indeed the differential genes are from the cited study, the authors should re-do the analysis using their own data, so as to correctly match the HiC and capture HiC data.

We have made it clearer in the text (e.g. ‘Data availability’) that the genes which are described as differentially expressed in our study are the results of re-analysing data from the study by Hoare et al. 2016, which is a study performed in our laboratory (now lines 128-130). The identical RIS cell model was used to generate the expression data and the Hi-C and capture Hi-C data. Similarly, the studies by Parry et al. 2018 and Chandra et al. 2012 were also performed in our lab using the same model and we re-analysed the ATAC-seq and ChIP-seq data mentioned in these studies to match the analysis of the ChIP-seq data generated for this study (Methods, original lines 538-540, now lines 664-668).

9) For the statement “The complex rewiring was exemplified by the IL1 and MMP loci...”, it is not shown in the figure 2a and extended figure 4a, how the IL1 and MMP loci being “rewired” has what effects on the genes. For making this claim, the global relations between this “rewiring” should be analysed, also especially with the given specific examples. For instance, an alternative interpretation of the MMP10 example (extended figure 4a) associates with 5 decreased interactions with enhancers, and 3 increased interactions with enhancers, while gene expression is up-regulated. One could hypothesise that the 3 enhancers are

stronger enhancers than the 5 combined. MMP1 loses interactions with 2 out of the 3 “stronger” enhancers (extended figure 4a), and yet the expression still increases.

While there is no doubt that a lot of the enhancers are “interacting” with the different genes, it is very unclear how many of these interactions or rewiring are actually significant or important for changing gene expressions. To make any claims about examples, the authors should put this into context of a global analysis.

Also, the authors did not exclude increased gene expression due to for example increased transcription start site chromatin accessibility or activation histone mark for instance.

To really establish the link between EP changes and gene expression further analyses and/or experimentation on specific loci, are required to characterise and disentangle the mechanisms of this “complex rewiring” as the authors have put it.

The scope of our analysis was to identify interaction changes between growing and RIS which co-occur with gene expression changes and to determine the pattern of interaction changes, which correlates with loop disruption. Indeed, this was a global analysis, and based on this genome-wide analysis, we exemplified two senescence-relevant regions. The actual number of the significantly altered EP pairs identified using the given criteria (enhancers with sizes greater than 7.5 kb and bin sizes smaller than 30 kb) was stated in the text as “719 EP changes genome-wide from Hi-C data, involving 553 differentially expressed genes” (original Extended Data 5a, new Supplementary Fig. 5a). The reviewer’s interpretation is well aligned with ours that, as suggested later in the manuscript, transcription-dependent new cohesin peaks in RIS cells correlated with the altered EP-interactions, which might lead to more efficient EP combinations (e.g. in *IL1* and *MMP* loci).

We agree it is interesting to consider other factors, such as active histone marks, for gene activation. Related discussion can be found in our response to Reviewer 2-point #2. We have examined the relationship between EP changes, gene expression, and Histone H3K27ac (on enhancers), and found a significant positive correlation between increased EP interaction and H3K27ac intensity or gene expression (adj. $p=6.383e-6$). This information is now in the main text (line 203-206). It is difficult to dissect between alterations in EP interaction and epigenetic marks, but this is an interesting question for future work.

10) Based on gene ontology enrichment of up-regulated genes in inflammatory terms and down-regulated genes in cell cycle terms, the authors concluded “This suggests that the two senescence hallmarks, the SASP and proliferative arrest, are controlled through the rewiring of the EP network.”. However, as pointed above, the actual relation between the rewiring of enhancer promoter pairs remain unexplored in this study, and therefore this conclusion cannot be drawn. The results could only allow the authors to conclude that gene expression changes in the RIS condition indeed agrees with the senescence phenotype as shown in GO enrichments. Further experimentation on specific loci using methods such as CRISPR + capture HiC + qPCR should be required to establish a clear mechanism between the EP and gene expression changes.

We agree this is an important point. As suggested by the reviewer, the CRISPR approach is interesting. Unfortunately, CRISPR-mediated deletion of a large genomic fragment in a

homogenous population is still challenging in primary cells. We have clarified that our data are focused on a global correlation and we have toned it down as below.

“While further experimental validation is required, our data suggest that the two senescence hallmarks, the SASP and proliferative arrest, might be controlled through the rewiring of the EP network.”

This point is further discussed in our response to point 11 below.

11) Based on the co-occurrence of cohesin binding and EP contacts at the same site at the IL1 locus, the authors concluded that “Collectively, these data suggest that the de novo cohesin peak contributes to the formation of new loops in the IL1 locus and that within each loop domain, EP pairs might preferentially contact (Fig. 3g).”

However, the data only showed the co-occurrence of EPs and cohesin loops, and the authors did not demonstrate that cohesin indeed contributes to the formation of the loops at the IL1 locus. Experiments to demonstrate this causal relation should be performed in order for this conclusion to be made.

CRISPR-targeting cohesin islands would be most relevant in this case but they often span gene bodies and their deletion would directly ‘knockout’ the gene expression, thus interpretation would be difficult. We are currently considering using CRISPRi epigenetically targeting the specific ‘promoter’ of a cohesin island-positive gene and test, as a consequence of the cohesin island’s deletion (as we expect), whether it affects new loops. Our preliminary results suggest that this approach would be more suitable for a bulk heterogeneous cell population (e.g. primary cells) compared to the standard CRISPR approach. Unfortunately, this is still ongoing, and, considering the strong genome-wide correlation between de novo cohesin islands and new loop formation, we feel this is beyond the scope of this study. We hope to do this in the future.

12) Based on the results in figure 4 and extended figure 9, the authors concluded that “cohesin-mediated loop alteration at the IL1 locus (as in RIS) facilitate transcription of genes in this locus”. However, it would appear that, based on their experiments using DRB, transcription would facilitate the formation of cohesin loops. The causal effect between transcription and cohesin loops at this particular IL1 locus has not been demonstrated and should be determined experimentally. For instance, by removing the cohesin binding sites using CRISPR and determine how gene expression is affected. Or globally by overexpressing cohesin to determine if transcription is affected in order to establish the mechanisms.

Line 279 As previously mentioned, the authors did not show an association between altered EP and gene expression, and should be determined by further experimentation as mentioned above.

This is an important question – what exactly is the relationship between transcription and cohesin island-loops? As the reviewer indicates, the DRB experiment suggests that cohesin island accumulation during RIS is transcription dependent. This effect is universal in the genome-wide analysis. To further support this, we have analysed the specificity of this effect:

a new Figure (Supplementary Fig. 10c) shows that, although the DRB treatment diminished cohesin islands, it did not affect other cohesin peaks that colocalize with CTCF (see our response to **Reviewer 3-10**). On the other hand, the potential impact of new loop formation (that is associated with cohesin islands) on the activation is more correlative. This is based on the increased EP interaction within the new loops. We discussed an interesting possibility that cohesin island-associated new loop formation and gene activation might reinforce each other (a feed forward mechanism). We have clarified this is a model, not a conclusive statement.

As discussed in our response to point 11 of this reviewer, removing cohesin island regions would be difficult, since it would directly affect gene expression. Also, CRISPR-mediated deletion of a large genomic fragment in a homogenous population is still challenging in primary cells. Although it would be an indirect approach, we are currently planning to epigenetically silence specific cohesin island-positive genes via promoter-CRISPRi with using RNAi as a negative control and examine gene expression in ‘other’ neighbouring genes. However, it is still an ongoing challenge and we are hoping to tackle this in follow-up studies for individual genes of interest.

Minor comments:

a) The correlation between SMC3 and RAD21 genome wide chip seq signal should be shown in supplementary, including p-values.

We have added a supplementary figure with all the pairwise correlations between every SMC3 and RAD21 sample and included the p-value description (new Supplementary Fig. 6a). The correlation tests were Pearson’s product-moment correlation and all p-values were below $2.2e-16$ and thus, each pair of samples shows significant correlation.

b) we are unable to find the relevant section in the methods that is supposed to explain figure 3b regarding CTCF and cohesin binding changes.

CTCF and cohesin significant binding changes were calculated with THOR (a similar tool to diffbind) as described in the Methods line 475, now line 685. We further clarified that THOR was used to generate the data behind the tracks in all Figures and behind the number of changes in Fig. 3a, b (now also Fig. 3a, b). In new Fig. 3b (original Fig. 3b), we show where those changes occur on the genomic regions spanning the loops determined by Rao et al. 2014 (close to the loop ends or inside the loops). Each loop segment is represented radially and scaled so that all the segments have the same length and chromosomes are segregated by white space. We have added this clarifying information in the Data Visualization Methods section of the manuscript (new lines 628-632).

c) The authors should show an example of the data of where the cohesin “anchor” is in relation to the CTCF in the extended figure. Also describe the parameters of how cohesin anchor is defined.

We have mentioned in the main body of text that we used the loop definition from Rao et al. 2014 about growing IMR90 loops and thus the anchors used are from this set of loops. We

checked that those loops are indeed bound by CTCF and cohesin from our ChIP-seq data in IMR90 cells as described in the study by Rao et al. 2014 and added this information in the revised manuscript (new lines 266-268).

d) The method description is missing related to figure 3c, regarding CTCF and cohesin loops

We have described the interaction aggregation method in the Interaction Neighbourhood Analysis Methods section (original lines 523-531, now lines 651-659). This is a re-implementation of the widely used APA (Aggregate Peak Analysis) described in Rao et al. 2014. We now emphasised this in the main body of text and provided additional mechanistic details (new lines 280-283). The loops we used in our analysis were defined in the study by Rao et al. 2014 who use a deeply sequenced Hi-C dataset in the same cell line we use, the IMR90 cells.

e) Figure 1: The authors did not clarify whether the illustrated figures are representative of both replicates.

We have further clarified in the now Fig. 1a legend that the visualisation of Hi-C matrices in each condition was performed using the Hi-C interaction counts across all the replicates, aggregated as described in lines 487-490 in the Methods section regarding data visualisation (now lines 608-611).

Reviewer #2:

The authors performed a study of the 3D rewiring of chromosomes in oncogenic HRAS-induced senescence by combining Hi-C and capture Hi-C, and associate it with cohesin redistribution. This is an important and topical study shedding new light on the chromosome conformation changes driving the expression of SASP genes in oncogene induced senescence. However, some of the results presented should be better clarified and validation is partially missing.

Major Revisions:

1. The authors should better clarify how enhancer regions are defined. Were histone marks or enhancer RNA's considered? If so, please emphasize this data more. It is mentioned that EP were considered only if enhancers were between 7.5 KB and smaller than 30 KB. However, this does not address how enhancer are defined in this study.

Enhancer regions were defined as H3K27ac peaks which show overlap with H3K4me1 and ATAC-seq peaks and no overlap with gene promoters. Furthermore, similar to the ROSE algorithm of determining super-enhancers and to the enhancer definition in the study by Tsdemir et al., we summarise peaks which are less than 12.5 kb apart into a single enhancer. This was mentioned in line 453 (now line 570) in the Methods section. We have elaborated on this description for further clarity in the Results section of the manuscript (new lines 172-174).

2. Are the EP pairs found in senescence due to the formation of novel enhancers? Please discuss this more in the main body of text.

Very few EP interactions (83) involve a novel enhancer defined as an enhancer with high H3K27ac in RIS and overlapping a region with significant increased H3K27ac (as calculated with THOR) and close to no H3K27ac in Growing. However, this may be because changes in levels of H3K27ac (with pre-existing H3K27ac) are more prominent, rather than entire new enhancer formation. We checked the correlation between the change in H3K27ac over an enhancer and the expression log-fold change of its associated gene, as well as the EP interaction log-fold change and found they are positively correlated. We now have added information in the text (new lines 203-206) regarding the correlation between the EP interaction/gene expression log fold-change and the changes in H3K27ac.

3. Line 279: It would be informative to do a motif analysis for TFs in loops/enhancers that forms in response to TNF-alpha treatment compared to those that form in senescence.

	Common Enhancers	RIS Enhancers	TNFa Enhancers
1	AP-1 	ATF3 	ATF3
2	BATF 	AP-1 	AP-1
3	FOS 	BATF 	FOS
4	JUNB 	FOS 	FRA1
5	ATF3 	FRA1 	JUNB
6	FRA1 	JUNB 	BATF
7	FRA2 	FRA2 	FRA2
8	FOSL2 	FOSL2 	FOSL2
9	ETV2 	JUN-AP1 	JUN-AP1
10	JUN-AP1 	ERG 	NFKB-p65
11	ERG 	ETV2 	ERG
12	BACH2 	ETS1 	NFKB-p65-Rel
13	EWS:ERG 	BACH2 	EHF
14	PU.1 	EWS:ERG 	EWS:ERG
15	SCL 	EHF 	BACH2
16	ETS1 	ETV1 	GA-repeat
17	ELF4 	ELF3 	ETV2
18	ETS1-distal 	ELF4 	TEAD
19	NFKB-p65 	ETS1-distal 	TEAD1
20	ELF5 	SCL 	TEAD4

Fig. R5: Top 20 motifs enriched using HOMER against known-motifs in each set of enhancers: common between RIS and TNFa, specific to RIS and TNFa treatment,

In concordance with Jin et al. 2013, we find no new loops forming or any significant interaction change with TNF-alpha treatment by re-analysing their full Hi-C data. However, increase in H3K27ac is reported with TNF-alpha treatment. We defined the sets of “TNFa specific Enhancers” and “RIS specific enhancers” as the enhancers which show increased H3K27ac in each condition (defined by differential binding analysis with THOR) and performed motif analysis on the two sets, as well as on the enhancers common in the two

conditions, using HOMER. The top motifs enriched in each set are highly similar with only slight re-ordering. The top motifs are ranked by adjusted p-values (Benjamin).

4. No validation is presented for any of the interactions discussed in the manuscript. For example, the dissociation of *NRG1* from the local chromatin environment or the loop formation of *IL1B* could be verified with microscopy.

We agree this is important. As suggested, we have performed DNA-FISH experiments to show the dissociation of *NRG1* (gene-probe) from the H3K27me3 TAD (TAD-probe) it belongs to, as well as *HMGA2* from its neighbouring H3K27me3 region. The location of those probes can be found in new Fig. 1b (*NRG1*) and new Supplementary Fig. 2a (*HMGA2*).

Representative FISH images (*NRG1*) and quantification of the distance between of gene signals (Green) and corresponding TAD signals (Magenta) both in Growing and RAS-induced senescence (RIS) conditions are shown below (new Fig. 1d, e). Consistent with our model, the gene-probes, but not the TAD-probes, tended to exhibit a more relaxed pattern, as seen in the representative FISH image below (see new Supplementary Fig. 2d for *HMGA2*). We have now described those experiments in the main body of text (new lines 155-162), as well as in methods (new lines 461-478).

Note, even if the base-pair distance between gene-probes and TAD-probes is quite small (~600 kb), compared to the probe size (~170 kb), we captured the significant increase in the distance between gene- and TAD-probes during RIS.

Fig. R6: Quantification of average distance between FISH signals corresponding to the *NRG1* or *HMGA2* gene (Green) and its neighbouring H3K27me3 region (Magenta) in both growing and RIS (new Fig. 1d, e).

Minor Revisions:

1. Please specify the resolution for the contact matrices when comparing biological replicates and calling TADs in the main body of text.

We now added this information in the main body of text: we called TADs at 40kb resolution and we compared biological replicates at every resolution between 10kb and 200kb (new line 123, legend of Supplementary Fig. 1a,b).

2. Line 80: TAD borders are compared between conditions. What window size is used to compare TAD borders? Please specify in figures and in main body of text.

We compared the exact location of TAD borders determined at 40kb resolution, using normalised mutual information (we now added details about this in the main text – new line 123 as well as TADs methods section, new lines 540-546). We did not compare borders which are within 40kb of each other because of the high level of agreement observed even with the stringent comparison of the exact borders.

3. Line 175: Why would CTCF binding increase in senescence if cohesin binding is decreased? Are there any other known roles of CTCF binding that are independent of cohesin formation?

This is an interesting question. In this study, we mainly focused on de novo cohesin binding, since the alteration of CTCF does not appear to explain differential interactions in our Hi-C data and the proportion of CTCF changes relative to the total number of CTCF peaks is rather small (4% of the number of growing peaks). We also checked the correlation between CTCF increased binding and gene expression or increased accessibility and found no obvious trend. The increase of CTCF binding co-localises with gene bodies and sometimes interior of the loops reported by Rao et al. and rarely with promoters or loop ends. We would like to follow up studying the relevance of increased CTCF binding in senescent cells in the future.

4. Line 195: This sentence is confusing, please clarify. Are you saying that the regions where CTCF decreased in binding in RIS did not result in complete loss of CTCF at these sites?

The reviewer is correct (we assume the reviewer means cohesin, not CTCF). Our quantitative ChIP-seq analysis identified cohesin peaks with differential binding between control and senescence, but the increased peaks (in senescence) are mostly de novo (no detectable peaks in growing control). We clarified in the manuscript that the incomplete loss of binding refers to cohesin, not CTCF (new lines 298-300).

5. Line 408: HiC protocols generally call for 2-5 million cells per replicate. Is '50 million' a typo?

50 million cells is not a typo, a lot of material is needed for Hi-C and especially capture Hi-C experiments in order to not have to over-amplify the library by increasing the number of PCR cycles, which results in lower complexity and high number of duplicates. The study by Schoenfelder et al. 2018, JoVE, PMC6102006 states that one should start with at least 20 million cells at the very minimum. For example, in the study, Schoenfelder et al. 2018, Nat. Comm, PMC6180096, we used 30-40 million cells. We believe that the recommendation for high-quality Hi-C libraries currently is 30-50 million cells using a restriction enzyme with a 6

bp recognition site (such as HindIII in this study), and 2-5 million cells when using a restriction enzyme with a 4 bp recognition site (such as DpnII or MboI). For example, DpnII has ~9 times more restriction sites in the human genome than HindIII, and as a result DpnII Hi-C libraries are ~9² more complex compared to HindIII Hi-C libraries; therefore a lower number of starting cells is required for DpnII Hi-C libraries.

6. Line 424: How strong is the correlation between H3K4me1 signal and the principal component? Please show data, perhaps a meta plot.

We compared the A/B compartment score calculated in 100kb bins with the number of peaks per bin of both H3K4me1 and H3K27ac. Correlation values between H3K4me1 Growing peaks and RIS peaks, respectively, and the AB score were both equal to 0.72. Moreover, correlation values with H3K27ac peaks were 0.58 and 0.61 in Growing and RIS, respectively. The values differ slightly between Growing and RIS because there is an increase in H3K27ac peaks in RIS, whereas the AB score is very similar between Growing and RIS. We now have added a figure (new Supplementary Fig. 1d) depicting those correlation values.

7. Line 463: The rigor here is appreciated, but why would these interactions be "false positives"? Is this due to the limited resolution in Hi-C libraries as compared to capture Hi-C libraries?

Yes, this is due to reduced sampling of the total number of reads in Hi-C compared to capture Hi-C libraries, as well as a consequence of using larger bin sizes in Hi-C (we tested 10-100kb) than the capture Hi-C differential analysis which was performed at *HindIII* (individual restriction enzyme fragments) resolution, which is ~4kb.

8. Please discuss further why cohesin islands are more present in RIS conditions when compared to proliferating controls.

This is a critical point. While cohesin island-like phenomenon has been suggested in wild-type yeast, it has only been reported in *Ctcf-Wapl* double KO cells in mammalian cells. Consistently, we failed to find cohesin islands in normal fibroblasts. We now extended the Discussion section regarding cohesin islands specificity to RIS, particularly focusing on differences between RIS and *Ctcf-Wapl* double KO cells.

9. Methods-Cell Culture: Please give details on TNF alpha treatment. What marker genes were tested to verify the efficacy of the treatment. How long did the treatment last? Was FBS concentration reduced? Is there TNF-alpha in the FBS already?

We used 10 ng/mL concentration of TNF α for 1h treatment before harvesting as used in the previous Hi-C study study (in Jin et al. Nature 2013, 503, 290). For a direct comparison with the senescence experiments, we used the same culture condition (10% FBS). We measured *IL1* expression (qPCR) as well as RELA localisation (IF), indicating a highly synchronous activation of NF- κ B signalling (nuclear localisation of RELA) with strong upregulation of both *IL1A* and *IL1B* mRNAs as shown below. These results can now be found in new Supplementary Fig. 7.

Fig. R7: Left: Nuclear localization of RELA with TNF α treatment; Right: Up-regulation of IL1A and IL1B with TNF α treatment relative to growing control, tested with qPCR.

Reviewer #3:

In this work, the authors examined the changes in chromatin structure that are associated with the changes in expression that occur during oncogene-induced senescence in human diploid fibroblasts. Using in situ Hi-C, together with previously obtained genome-wide data, they first identify 102 up-regulated genes that dissociate from H3K27me3-enriched regions and suggest that this may be a “relatively common mechanism of gene activation” during this senescence (p. 4). They then combined in situ Hi-C with capture Hi-C, and previously obtained data, to identify 1004 enhancer-promoter (EP) interactions that change significantly during this senescence and that are associated with differential expression. The genes associated with these altered EP interactions were found to be enriched for “inflammatory” and “cell cycle arrest” genes, which led the authors to conclude that the two “hallmark” features of senescence, the senescence-associated secretory phenotype (SASP) and proliferative arrest, “are controlled through the rewiring of the EP network” (p. 5). Finally, as the proteins CTCF and cohesin are believed to be directly responsible for the formation of chromatin loops that underlie many EP interactions, the authors used ChIP-seq to characterize their genomic location and extent of binding during senescence. They found changes especially in cohesin binding that are associated with alterations in many loops. Yet perhaps most intriguingly, they identified 574 extended regions of de novo cohesin binding that resemble previously observed “cohesin islands” in (CTCF and Wapl) double knockout (DKO) mouse embryonic fibroblasts. By resemblance, the authors note an enrichment at the 3' end of expressed genes, an elongated shape, as well as a loss of binding upon

transcription inhibition, but could have (probably) also mentioned a more extended binding region than is typically observed associated with CTCF. They also note similar findings upon re-analysis of previously published monocyte-to-macrophage data. With this, the authors conclude that these cohesin islands do indeed “occur in physiological contexts in mammalian cells” (p. 12), where they could play a role in gene regulation during cell fate determination.

Overall, this is a very good piece of work: it is well written, thoroughly analyzed, and contains a copious amount of data that largely supports their main conclusions. At the moment, there is considerable interest in understanding the biological processes underlying senescence, as well the link between chromatin structure and gene expression. This work is a useful, and perhaps significant, contribution in both areas. My prime concern is the absence of analysis that quantitatively supports the conclusions that these identified structural changes are prominent mechanisms by which the senescent phenotype is effectuated.

Specifically, my major concern is:

1. The authors should mention the number of genes, overall, that are differentially expressed, as well as up- or down-regulated. It is otherwise difficult to judge if indeed the 102 that dissociate from the H3K27me3-enriched regions do reflect evidence for a “common” mechanism of gene regulation in senescence. Similarly, a quantification of the total number of differential expressed genes that are associated with the two senescence hallmarks is also necessary to determine if the fraction associated with changed EP interactions is significant enough to justly render the re-wiring of this network as “controlling” these two hallmarks.

We agree this is a valid point. By naming it a “common” mechanism, we intended to highlight that the three-dimensional detachment from H3K27me3 regions is not a rare event. However, this term is vague. Indeed, the reviewer made a very good point: the 102 genes are only 5% of all upregulated genes during RIS and we cannot justify the word ‘common’. Therefore, we removed this statement (i.e. removed: “and that release from such domains appears to be a relatively common mechanism”). We thank the reviewer for pointing this out. We also added information regarding the percentage of all differentially expressed genes corresponding to the genes involved in EP interactions (new line 209).

Other concerns are:

1. While some of the observations here do resemble the “cohesin islands” previously observed in DKO cells, there are also significant differences. In particular, those observed here are found to be associated with genes that are specifically enriched for pathways important in senescence, while no such selectivity (based on gene function) is apparently present in the DKO cells. In addition, while those in the DKO study are found to redistribute to the transcription start-site (TSS) upon transcription inhibition, those observed here do not (Fig 4b). Those observed here also seem to extend across the entire gene, in contrast to those in DKO cells. The local minimum specifically at the TES in Fig 4a and Extended Data Fig 9b is also unlike the corresponding profile from the DKO study. Also, one might predict that CTCF binding within the MMP1 gene would prevent cohesin island formation (according to the model presented in the DKO study), but it clearly does not here (MMP1 in Extended

Data Fig 8). Finally, all of the cohesin islands in the DKO study were located downstream of actively transcribed genes, whereas with 574 islands overlapping near the 3' ends of 343 genes in this study (p. 10), I believe that would leave 231 islands that are not associated with the 3' ends of genes. It is thus not clear that the mechanism underlying their formation is the same as in the DKO cells. As such, there is no evidence presented here to justify the phrase, "transcription-driven" in the title. In the DKO cells, it was the repositioning to the TSS upon transcription inhibition and the changes in peak shape between converging genes as a function of the relative extent of expression that were consistent with a "transcription-driven" mechanism. Perhaps "dependent" instead of "driven" would be more appropriate in the title. In addition, the authors may wish to include some discussion of these differences with the DKO cells, as they may help future efforts to uncover the mechanism.

The reviewer raises a number of interesting and important points. First, is there any 'selectivity' of cohesion islands for specific genes? This remains to be elucidated but the strong correlation with transcription activity implies in both the DKO study and our study suggests that the cohesin islands observed in senescence were enriched for genes important for senescence mainly because cytokines and other genes belonging to the Senescence-associated secretory phenotype (SASP) are very highly expressed and they constitute a substantial percentage of all highly expressed genes. Other genes known to be highly expressed, such as *ACTB* (b-actin) also present a cohesin island. Importantly, *ACTB* is highly expressed in growing cells as well, but it has a cohesin island only in the senescence condition. Therefore, we speculate that highly active transcription is required but not sufficient for cohesin islands formation. It is tempting to speculate that the transcription-dependent cohesin accumulation and gene activation reinforce each other, eventually contributing to forming the phenotype specific gene expression profile.

As this reviewer points out, RIS-associated cohesin islands are often observed in gene bodies, not only at 3'-ends. Indeed, they appear even outside genes and overlapping enhancers. We have added a figure depicting their localisation preferences and updated all the figures involving cohesin islands to take all of them into account, not just the ones overlapping 3'-ends (new Supplementary Fig. 9d). RIS-associated cohesin islands also tend to be narrower than those reported in the DKO study. We have now included a discussion on the difference between the cohesin islands in DKO cells (Busslinger et al.) and senescence (this study).

Interestingly, Busslinger et al. mention that cohesin islands, although less pronounced, are also observed in the *Wapl* single KO condition. It is possible that the presence of CTCF in our condition might contribute to cohesin accumulation on gene bodies, rather than solely accumulating at 3'-ends of genes.

The presence of cohesin islands at multiple genic sites is possibly due to the relationship between cohesin and Pol II which has been mentioned in the study by Busslinger et al., as well as by others (e.g. Schaaf et al. 2013). We performed Pol II ChIP-seq to confirm this relationship between cohesin islands and Pol II and indeed noticed that the formation of cohesin islands is closely correlated with Pol II signal. We have added this information in the

text (new lines 373-376), as well as in new Fig. 5c (as well as signal tracks for Pol II in Fig. 4b and Supplementary Fig. 8b).

Altogether, the most critical feature of cohesin islands are perhaps their association with transcription activity. Nevertheless, our data also suggest highly active transcription alone is not sufficient for driving cohesin islands, thus we have now changed the title to “dependent” rather than “driven” as suggested.

2. The PCA results in Extended Data Figure 1 indeed show that the “Growing” replicates cluster together, but strangely, the RIS replicates do not. It is not clear that one could conclude from this calculation that the RIS replicates are indeed consistent with each other, although the results from HiCSpector look good. Perhaps the authors could provide some additional discussion for the PCA results or use another tool, such as HicRep, to further substantiate agreement between the RIS replicates.

Yardımcı et al. 2019 (Genome Biology) performed a comparison between methods for assessing agreement between Hi-C samples and highlighted HiCSpector and HiCRep as superior to ad-hoc methods such as correlation coefficient. However, these methods have not been adapted to highly sparse data such as capture Hi-C matrices. This is the reason why we performed PCA on filtered, library normalised counts. We also used PCA to check whether the largest source of variation is the main effect studied, the changes between Growing and RIS. Moreover, the two RIS replicates are close to each other in PC3 vs PC4 (below, left) and the variation explained by PC2 and PC3 is similar.

We also used HiCRep to check agreement between replicates and we now have included only the results from HiCRep and HiCSpector to avoid further confusion in new Supplementary Fig. 1a-b (below, right). We thank the reviewer for this suggestion.

Fig. R8: Principal component analysis: PC3 vs PC4 values corresponding to growing and RIS biological replicates

Fig. R9: HiCRep Scores representing agreement between pairs of biological replicates in growing and RIS

3. Were the EP interactions from the Hi-C data determined from only one replicate? If so, how do they compare with those determined from the other replicate?

All replicates (3 Growing and 2 RIS) were used to determine significantly changing interactions (clarified in the text – new line 131 - and in the legend of Fig. 1). We used diffHic (Lun et al. 2015, line 441, now line 558 in Methods section) which performs differential interaction analysis using an adaptation of the statistical strategies used by edgeR for differential expression analysis. The EP interactions are a subset of the interaction changes determined, defined as interactions where one anchor overlaps an enhancer and the other overlaps a gene promoter.

4. By the authors' own criteria of filtering the Hi-C EP interactions, I believe that Extended Data Fig 4b shows that enhancers larger than 5 kb would have been a better choice than those larger than 7.5 kb.

The two filtering strategies: enhancers larger than 5 kb and 7.5 kb are similar in the two plots. The 5 kb allows for more interactions to be identified which are in both the Hi-C and the capture Hi-C, whereas the 7.5 kb one identifies slightly fewer interactions. Thus, the 5 kb cut off has slightly higher sensitivity. However, the 5 kb option also has a higher “false positive” probability, i.e. positive in Hi-C but not in capture Hi-C (left panel of the original Extended Fig. 4b, now Supplementary Fig. 4b). We chose the safer option of enhancers larger than 7.5 kb.

5. On p. 6, the authors describe the formation and loss of many loops within the IL1 cluster based on their cHi-C data (Fig 2f), while on p. 8, line 197, the authors mention that “The genes and enhancers studied in the IL1 locus belong to a single loop identified in IMR90 cells (ref 8). This is consistent with enhancer sharing between these genes...”. How could the presence of a single loop in IMR90 be consistent with enhancer sharing that the authors articulate occurs with the formation and loss of the many loops that are shown in Fig 2f?

We meant the promoters of *IL1A*, *IL1B* and *CKAP2L* belong to the same loop, which does not contain any sub-loops in Growing and that this loop breaks into two smaller ones during senescence (RIS). We have now clarified in the text that we meant that the genes belong to the same loop rather than a single loop (now line 300).

6. I am a little confused by the following numbers: the author state that “3407 (out of 7,647) loops showed changes in cohesin binding, mostly decreases (80%)” (line 180, p. 7), which corresponds to 2726 loops. But later (line 187), the authors mention that “Loops with cohesin loss at one or both ends (1,827 loops)...”. What is the reason for the difference of nearly 900 loops?

We apologise for the imprecise wording, it is due to using a more robust definition of loops with cohesin losses when profiling the average interactions. We noticed that the cohesin/CTCF peak associated with the loop end is not always exactly on the loop end (10kb region) reported by Rao et al. This can be seen in new Figure 3b (original Figure 3b), where

some cohesin losses fall near the loop end but also in the overlaps of the loop ends with both CTCF and cohesin (90% fall within 10kb of the reported loop end). This is expected due to variability in the data, as well as the original variability in the process of loop calling, which may call loops in the bin next to the actual loop end. The 80% loops in question show cohesin losses at least at one end **within** 10 kb of the loop anchor, so it takes into account this variability. Some of those loops also show a cohesin decrease at one end and an increase at the other and the resulting interaction changes are mixed and difficult to interpret. Therefore, to check for a trend in interaction changes, we used a more precise definition for the subset of loops with cohesin losses so that the cohesin decrease falls precisely on the anchors reported by Rao et al. with no cohesin gains at the other end. This subset consisted of 1,827 loops.

In order to address this potential confusion, we have now used a more straightforward characterisation, which only reports the most stringent changes in cohesin falling exactly on the loop anchor. Note that the trends in Fig. 3c (original Fig. 3c) are essentially the same regardless of the subset of loops used (the stringent or the one which considers cohesin changes within 10kb of the loop end). Due to the higher stringency in the loop definition, more loops may be disrupted in senescence but for easier comprehension of the reader, we think it is best to use the more straightforward characterisation. We also added the figure below (now in Supplementary Fig 6b) in order to clarify the contribution of cohesin alterations compared to CTCF and the overlap between different subsets of potentially altered loops, as well as provide an estimate of potentially affected EP interactions by all types of CTCF/cohesin alterations of the loops they reside in.

Fig. R10: Number of potentially disrupted loops in terms of gain/loss of cohesin/CTCF at loop ends or inside the loops

7. With respect to Fig 3g, the new cohesin binding site at IL1B was shown to interact more with the anchors of the loops (Fig 2d,f). That is, with the anchors to the right in Fig 2f. There are no anchors to the left in this image and there is no evidence for increased contact between the cohesin binding site in the Hi-C with any putative anchors on the left. Yet, I believe Fig 3g depicts contact between the cohesin site and an anchor on the left.

Our interpretation of the contact changes in new Fig. 4e (original Fig. 3g) is that the new cohesin peak interacts the strongest with loop ends present on its left side, but also to its right. A new complex system of subloops appears in senescence, which we tried to simplify and explain with the cartoon as the formation of sub-loops within the loop which contains *IL1A*, *IL1B* and *CKAP2L* promoters. We hopefully clarified this in the text of the manuscript (new lines 313-315).

Fig. R11: IL1 locus: interactions of the new cohesin peak at the end of IL1B with flanking

8. I did not understand what the authors meant by (line 189, p. 8) “More stringently, 326 loops (Fig. 3c, right) were found to overlap with significantly reduced interactions during RIS, either from cHi-C or Hi-C (at 20 kb and 40 kb resolution)”. What does “more stringently” mean here? The figure legend indicates “Compare to all IMR90 loops with significantly decreased interactions during RIS (right)”, which did not help.

We apologise for the unclear description. In the original Fig. 3c right (now also Fig. 3c), we only considered loops with significantly reduced interactions. This visualisation method can capture the overall trend in the data. Then, in the left panel, we wanted to point out that, even if we include all loops, not just the ones which show significant changes, we see a general tendency of reduced interaction during RIS. In the revised manuscript, we have now clarified

this comparison and used the simplified definition of loops with cohesin losses at one of their ends or both, which we described in point 6.

9. In line 228, p. 9, the authors refer to Fig 4a as evidence for the accumulation of cohesin islands and cohesin binding that is correlated with gene expression, but this figure only shows an increase in cohesin binding, not the accumulation of cohesin islands. However, I believe that showing such an enrichment should be possible with their data.

Original Figure 4a was intended as an introduction prior to defining cohesin islands, as this was our initial observation regarding genes highly expressed showing cohesin accumulation past their 3'-ends. We now have added a figure (Supplementary Fig. 10a) showing that genes with cohesin islands (either at 3'-ends or gene body or promoter) are overall associated with higher expression than genes without cohesin islands.

10. Did transcription inhibition only reduce the binding at cohesin islands or also at “typical” cohesin binding sites (that is, those associated with CTCF)? Providing this information might also help future efforts aimed at resolving the mechanism.

Fig. R12: Left: SMC3 ChIP-seq signal of constitutive cohesin peaks (i.e. cohesin peaks with CTCF) with and without DRB treatment (new Supplementary Fig.10c); Right: Heatmap and profile of SMC3 with and without DRB treatment, as well as CTCF ChIP-seq

This is a good idea. CTCF-associated cohesin is not affected by transcription inhibition. We have now added a figure showing that the signal at those sites remains the same (new Supplementary Fig.10c).

11. A few concerns with the figures: in Fig 2f, the promoters for all genes, especially CKAP2L, are at the 3' ends of the genes; in Fig 3a, the y-axis is number, not frequency; in Extended Data Fig 1d, the blocks labeled HMGA2, IRAK3, and GRIP1 in the Hi-C map do not align with the genes in the track at the bottom of the figure; and the legend to Extended Data Figure 3b appears to be missing the words "active promoters" associated with the low/high histone modifications.

We thank the reviewer for his/her careful reading.

Fig. 2f (now Fig. 2f): As the reviewer correctly suggests, the transcription direction (arrows) in CKAP2L and IL1A was wrong. This error is now fixed.

Fig. 3a (now Fig. 3a): this is now fixed.

Extended Data Fig 1d (now Supplementary Fig. 2a): this is now fixed, re-plotted to add FISH probes used. Slightly different gene isoforms may be plotted for IRAK3 and GRIP1 because we used the gene body coordinates from GENCODE19 but in IGV Refseq genes are plotted by default.

Extended Data Figure 3b (now Supplementary Fig. 3b): We do mean all promoters of protein coding genes, not just the active ones.

REVIEWER COMMENTS

Reviewer #1 (Remarks to the Author):

Overall, the clarity of the manuscript has improved and the global analyses - many of which done already in the first version - are now clearly described.

The authors have partially addressed my points on the global analysis in the sense that they clarified that they did a global analysis. And for the global picture of the HiC data and the loops, I'm fully satisfied (and apologise for missing some of the points in the first round). However for the global analysis on the expression levels, I'm not fully satisfied by the answers: Even though the authors technically address my points in clarifying the analyses that were done, I'm still lacking the global picture in the presentation of the main figures and text. This is mainly related to the fact that many of the fold-changes that are now reported are rather small (see specific comments below), thus putting them in context of the full range of fold-changes observed (for anyone data type - RNA, H3K27ac, ATAC etc) is in my opinion essential to allow the reader to judge the effect sizes of the proposed model and mechanisms. Also, currently it almost seems the authors want to hide that this mechanism is only affecting a small number of genes, I think there is nothing to hide and the data should be presented upfront - either way it is an interesting observation. And given the comparison with the TNFalpha stimulation, nobody would expect that all the gene expression changes are driven by chromatin, and it is really important to show what fraction is impacted to what extent.

The specific suggestions are very simple analysis (see below)

Below two specific suggestions:

1) For the differentially expressed genes, NRG1 indeed shows a very significant log fold-change (4.6) while the gene in the second most differential TAD (HMGA2) shows a very modest log fold change of 0.7 (even if statistically significant). Presenting the data like this, raises the question if the second most significant gene only shows 0.7 fold-change, how small are the changes in the other 102 genes that the authors find? I would here really appreciate an honest figure about the effects, e.g. a volcano plot showing the differentially expressed genes and coloring the 102 genes they identified in a specific color. This would give the reader the ability to appreciate whether the findings are indeed a global phenomenon or a very specific mechanism for NRG1 - both of which are very valuable conclusions.

2) The same point applies to the next section "we identified 870 EP pairs that showed significantly altered interactions during RIS, involving 149 differentially expressed genes (Supplementary Fig. 4a)." The authors only show the identified pairs, and I think for the readers to make a conclusion it will be very important to get a good feeling about the extent to which such a mechanism is affecting genes to be able to put in context. Currently it seems the authors want to hide that this affects "only" 15% of genes - I think 15% is huge, given all the indirect effects that are expected to occur as a result of the up/down regulation of these genes. Showing the effect size of the genes (e.g. again with a volcano plot highlighting the affected genes), will also allow to appreciate the impact of EP changes - are these mostly modest changes in expression? Or are they among the strongest changes? No need to discuss this in detail, but the reader should be able to judge the data in a global context.

3) Similar comments apply to this sentence: "In terms of directionality, differences in H3K27ac binding over the enhancer in an EP interaction pair were positively correlated with the log-fold change of the interaction (0.24, p-value 9.353e-15) as well as with the gene expression log-fold

change (0.14, 6.383e-6)." These log-fold-changes are really small. It doesn't make them less interesting, but it is important to put them somehow in context of the overall observed log fold changes genome-wide.

Minor point:

In the sentence: "This was accompanied by a 145 significant increase in chromatin accessibility across the gene, as determined by differential binding analysis of growing and RIS ATAC-seq (Fig. 1b, bottom)." The authors still only show the ATAC-seq signal tracks. In the rebuttal letter they show the significance signal from their THOR analysis, which I would recommend adding to the figure. It is really difficult to judge from signal tracks whether or not a region is significantly different.

Reviewer #2 (Remarks to the Author):

The authors have addressed all of my points/concerns satisfactorily.

Reviewer #3 (Remarks to the Author):

The authors have satisfactorily addressed my previous concerns.

Reviewer #1 (Remarks to the Author):

Overall, the clarity of the manuscript has improved and the global analyses - many of which done already in the first version - are now clearly described.

The authors have partially addressed my points on the global analysis in the sense that they clarified that they did a global analysis. And for the global picture of the HiC data and the loops, I'm fully satisfied (and apologise for missing some of the points in the first round). However for the global analysis on the expression levels, I'm not fully satisfied by the answers: Even though the authors technically address my points in clarifying the analyses that were done, I'm still lacking the global picture in the presentation of the main figures and text. This is mainly related to the fact that many of the fold-changes that are now reported are rather small (see specific comments below), thus putting them in context of the full range of fold-changes observed (for anyone data type - RNA, H3K27ac, ATAC etc) is in my opinion essential to allow the reader to judge the effect sizes of the proposed model and mechanisms. Also, currently it almost seems the authors want to hide that this mechanism is only affecting a small number of genes, I think there is nothing to hide and the data should be presented upfront - either way it is an interesting observation. And given the comparison with the TNFalpha stimulation, nobody would expect that all the gene expression changes are driven by chromatin, and it is really important to show what fraction is impacted to what extent.

The specific suggestions are very simple analysis (see below)

Below two specific suggestions:

1) For the differentially expressed genes, NRG1 indeed shows a very significant log fold-change (4.6) while the gene in the second most differential TAD (HMGA2) shows a very modest log fold change of 0.7 (even if statistically significant). Presenting the data like this, raises the question if the second most significant gene only shows 0.7 fold-change, how small are the changes in the other 102 genes that the authors find? I would here really appreciate an honest figure about the effects, e.g. a volcano plot showing the differentially expressed genes and coloring the 102 genes they identified in a specific color. This would give the reader the ability to appreciate whether the findings are indeed a global phenomenon or a very specific mechanism for NRG1 - both of which are very valuable conclusions.

While we had attempted to address this in the paper in the form of correlations between interaction and expression log-fold changes, as well as H3K27ac changes, we agree that the link between chromatin interaction changes and magnitude of expression changes is important information. As suggested, we have included here and in the manuscript a volcano plot for all the differentially expressed genes with highlighting those 102 genes (purple) with less interactions with H3K27me3 regions. In addition, to better appreciate the magnitude of differential gene expression, we also have added a box plot for log-fold changes (absolute value) of these genes compared with all other differentially expressed genes. Indeed, the log-fold change values of the 102 genes are significantly higher in magnitude than those of other differentially expressed genes. We find this very interesting: although the values are quite

variable, the data suggest the extent to which expression of those 102 genes (reduced interactions with neighbouring H3K27me3 regions) is altered tends to be larger than the rest of differentially expressed genes. We have included this observation in the manuscript and we thank the reviewer for this useful suggestion.

Regarding *HMGA2*, its expression levels are often variable. We frequently compare gene expression results with other senescence RNA-seq datasets performed either by our laboratory or by others. The log-fold changes obtained for *HMGA2* in all datasets range between 0.4 and 1.3 and always have associated significant p-values even when the log-fold change is low. Indeed, as we showed in the previous version of the manuscript, we validated *HMGA2* levels by qPCR (log-fold change, ranged 1.2 – 2.4). It is also worth mentioning that the method used for detecting significant expression changes (glmTreat in edgeR) is quite conservative and the shrinking factors for the log-fold changes reported are likely higher than in the case of other less stringent methods. We apply this method because senescence is associated with a large number of expression changes which require more stringent testing approaches to avoid breaching the main assumptions behind the differential expression model.

Fig. R1 (corresponding to new Supplementary Fig. 2d):

Genes with increased expression and reduced contacts with H3K27me3 regions in RIS
Left: Volcano plot of all significantly differentially expressed genes highlighting the ones in the specified gene list; **Right:** Boxplots representing the magnitudes of the log-fold changes of genes escaping H3K27me3 regions and all other differentially expressed genes; significance testing performed using Student's t-test, *** $p < 0.001$

2) The same point applies to the next section "we identified 870 EP pairs that showed significantly altered interactions during RIS, involving 149 differentially expressed genes (Supplementary Fig. 4a)." The authors only show the identified pairs, and I think for the readers to make a conclusion it will be very important to get a good feeling about the extent to which such a mechanism is affecting genes to be able to put in context. Currently it seems the authors want to hide that this affects "only" 15% of genes - I think 15% is huge, given all the indirect effects that are expected to occur as a result of the up/down regulation of these genes. Showing the effect size of the genes (e.g. again with a volcano plot highlighting the affected

genes), will also allow to appreciate the impact of EP changes - are these mostly modest changes in expression? Or are they among the strongest changes? No need to discuss this in detail, but the reader should be able to judge the data in a global context.

Similar to point 1 above, as suggested, we have generated volcano and box plots for differentially expressed (DE) genes: the 688 genes (purple) that involve significant EP alterations in either Hi-C or cHi-C (the 15% of all DE genes) and the rest (grey) of DE genes. We agree the volcano plot puts those 688 genes in the global context. In addition, the magnitude of gene expression changes in those genes with EP alteration also tends to be larger than others. Undoubtedly other mechanisms are also involved but it is tempting to speculate that the dynamic alteration of distant chromatin contacts (point 1), including EP interactions (point 2), might amplify the effect of gene regulatory mechanisms. We thank the reviewer for these suggestions.

Fig. R2 (corresponding to new Supplementary Fig. 5c):

Differentially expressed genes involved in the Enhancer Promoter Network **Left:** Volcano plot of all significantly differentially expressed genes highlighting the ones in the specified gene list; **Right:** Boxplots representing the magnitudes of the log-fold changes of genes in the EP network and all other differentially expressed genes; significance testing performed using Student's t-test, *** $p < 0.001$

3) Similar comments apply to this sentence: "In terms of directionality, differences in H3K27ac binding over the enhancer in an EP interaction pair were positively correlated with the log-fold change of the interaction (0.24, p -value $9.353e-15$) as well as with the gene expression log-fold change (0.14, $6.383e-6$)." These log-fold-changes are really small. It doesn't make them less interesting, but it is important to put them somehow in context of the overall observed log fold changes genome-wide.

This question appears to be based on a misunderstanding and we apologise for the unclear description. The values mentioned (0.24 and 0.14) are 'correlation values' not 'log-fold changes'. To avoid any confusion, we have added the term (correlation value) before each number in the text.

Minor point:

In the sentence: "This was accompanied by a 145 significant increase in chromatin accessibility across the gene, as determined by differential binding analysis of growing and RIS ATAC-seq (Fig. 1b, bottom)." The authors still only show the ATAC-seq signal tracks. In the rebuttal letter they show the significance signal from their THOR analysis, which I would recommend adding to the figure. It is really difficult to judge from signal tracks whether or not a region is significantly different.

As suggested, we have added the significant THOR results below the signal tracks (Fig. 1b).

REVIEWERS' COMMENTS

Reviewer #1 (Remarks to the Author):

The authors have addressed all my points. Congratulations to the great work!